# Enhancement of electrocatalysis through magnetic field effects on mass transport

Priscila Vensaus [1,2,3,4], Yunchang Liang [1,2,4], Jean-Philippe Ansermet[2], Galo J. A. A. Soler-Illia [3] & Magalí Lingenfelder [1,2] ✉

Magnetic field effects on electrocatalysis have recently gained attention due to the substantial enhancement of the oxygen evolution reaction (OER) on ferromagnetic catalysts. When detecting an enhanced catalytic activity, the effect of magnetic fields on mass transport must be assessed. In this study, we employ a specifically designed magneto-electrochemical system and non-magnetic electrodes to quantify magnetic field effects. Our findings reveal a marginal enhancement in reactions with high reactant availability, such as the OER, whereas substantial boosts exceeding 50% are observed in diffusion limited reactions, exemplified by the oxygen reduction reaction (ORR). Direct visualization and quantification of the whirling motion of ions under a magnetic field underscore the importance of Lorentz forces acting on the electrolyte ions, and demonstrate that bubbles' movement is a secondary phenomenon. Our results advance the fundamental understanding of magnetic fields in electrocatalysis and unveil new prospects for developing more efficient and sustainable energy conversion technologies.

Ensuring a sustainable future and reducing carbon emissions heavily relies on renewable energy sources. Among these, electrochemical sources need to harness electrons to drive chemical reactions and store energy in the form of chemical bonds. HER (hydrogen evolution reaction), OER (oxygen evolution reaction) and ORR (oxygen reduction reaction) are essential electrochemical reactions for sustainable energy production, therefore playing a crucial role in a range of energy technologies. For example, HER is a key reaction in the production of hydrogen fuel through water electrolysis, which can be used to power vehicles and generate electricity with low or no greenhouse gas emissions. Similarly, OER is an important reaction in the production of oxygen gas from water, necessary for water electrolysis, which can also be used for industrial applications and in life support systems. Moreover, ORR is a critical reaction in fuel cells, which can be used to generate electricity by converting the chemical energy of fuels such as hydrogen or methanol into electrical energy. By understanding and optimizing the efficiency of these

electrochemical reactions, it is possible to develop new and sustainable energy sources that can help address global energy and environmental challenges, developing new approaches for enhancing the performance of electrochemical devices such as fuel cells, batteries, and electrolysers.

Numerous new catalyst materials and structures have been proposed to enhance the performance of the catalysts for these processes[1,2], while unconventional methods also introduce considerable improvements[3,4]. Among these methods, the application of a magnetic field in electrocatalysis is relatively less explored.

The study of magnetic field effects in electrochemistry can provide new insights into fundamental principles that can lead to the development of new tools and techniques for controlling (electro-) chemical reactions[5,6]. In particular, the study of the influence of magnetic fields on electrochemical reactions such as HER, OER, and ORR is of great interest due to the potential for improving the efficiency and selectivity of these reactions[3,7,8].

[1]Max Planck-EPFL Laboratory for Molecular Nanoscience and Technology, École Polytechnique Fédérale de Lausanne (EPFL), Lausanne, Switzerland. [2]Institute of Physics (IPHYS), École Polytechnique Fédérale de Lausanne (EPFL), Lausanne, Switzerland. [3]Instituto de Nanosistemas, Escuela de Bio y Nanotecnologías, Universidad Nacional de San Martín, San Martín, Buenos Aires, Argentina. [4]These authors contributed equally: Priscila Vensaus, Yunchang Liang. ✉e-mail: maggie@lingenfelder-lab.com

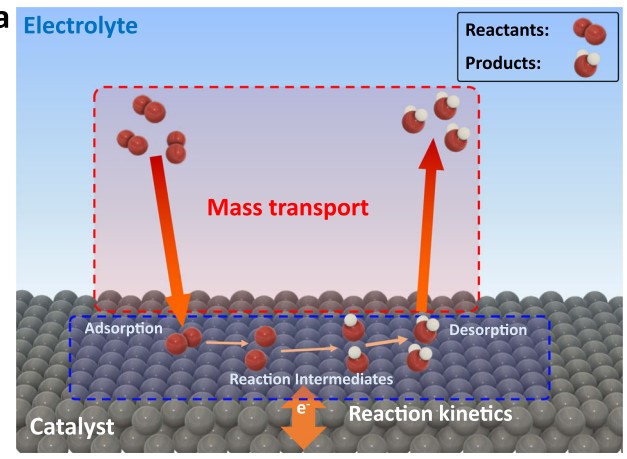

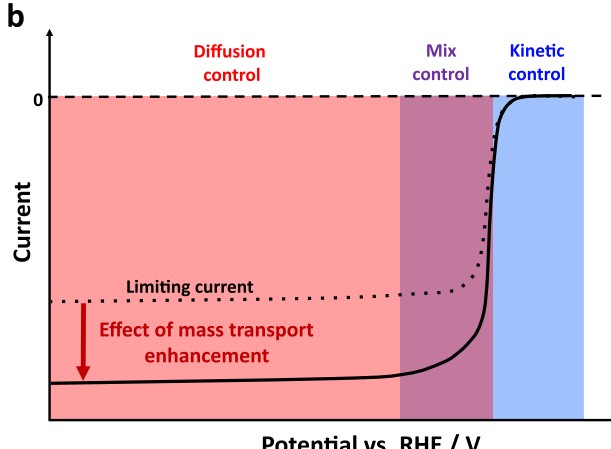

**Fig. 1 | Reaction kinetics and mass transport effects in electrocatalysis.**
**a** Illustration of an electrocatalytic reaction at the electrolyte/catalyst interface. The reaction comprises both mass transport processes and intermediate reaction steps at the catalyst surface. **b** Typical polarization curve of the oxygen reduction reaction indicating the kinetic, mixed, and diffusion-controlled regions, and the effect of enhancement in the mass transport.

In the context of the influence of magnetic fields on electrocatalysis, it is important to distinguish between two effects: kinetic and mass transport effects (Fig. 1). Kinetic effects, on the one hand, refer to changes in the reaction rate due to alterations in the reaction mechanism or kinetics, by modifying the rate of electron transfer or the activation energy required for the reaction to occur. Mass transport effects, on the other hand, refer to changes in the transport of species to and from the electrode surface. The magnetic field can alter the diffusion of reactants towards the electrode surface, or the transport of reaction products away from the surface. For example, in the case of oxygen reduction reaction (ORR), the behavior of the current as a function of the potential provides valuable insights into the contribution of these factors. When the potential in the cell is decreased, a negative current emerges due to the reduction of oxygen. This current typically decreases exponentially as the driving force for the reaction increases. The measured current is primarily influenced by the reaction kinetics in this regime. However, as the potential decreases further, a point is reached where the reaction proceeds faster than the arrival of reactants at the electrode surface. At this stage, the overall rate becomes limited by mass transport. This limitation is observed as a plateau in the current, known as the limiting current. Improving the mass transport of reactants, i.e., by stirring, can thus lead to a change in the limiting current. By separating the kinetic and mass transport effects, it is possible to gain a better understanding of the fundamental mechanisms underlying the influence of magnetic fields on electrocatalysis, and to optimize the design of electrochemical devices for sustainable energy production.

Garcés-Pineda et al.[3] have recently demonstrated a direct enhancement of electrocatalytic OER under a magnetic field and attributed this effect to changes in the magnetic (ferromagnetic or antiferromagnetic) properties of the catalysts. However, the effect of magnetic fields on the mass transport of chemical species from and to the electrode could not be completely ruled out. It has been noted that a magnetic field is capable of accelerating or redirecting the movement of gas-phase reaction products (bubbles), as well as directing or increasing electrodeposition currents[9,10]. The exact origin of the effect on the bubbles has been debated until recently. Monzon et al.[11] have concluded that a Lorentz force can be generated on gas bubble products (e.g., $H_2$ and $O_2$) under an external magnetic field due to the appearance of charges at the bubble surface. Moreover, these authors showed that the Kelvin force is likely not negligible when magnetic field gradients are strong at the magnetic catalyst surface[12,13]. Other authors attributed this effect to the formation of an ion current near the bubbles: as bubbles rise to the surface due to the buoyance force, adjacent ions are dragged by this motion, resulting in an ion current, which is then affected by a Lorentz force[14,15]. Some reports mention a change in bubble direction due to the interaction of paramagnetic $O_2$ with the magnetic field[16]. However, gas-phase products are not universal in general electrocatalytic reactions. Since mass transport in electrocatalysis mainly comprises diffusion, migration or convection of ionic species in the electrolyte, the effects of the magnetic field on the movement of these species require further investigations and evaluation.

In this work, we set up a magneto-electrochemical system that permitted us to study in detail the magnetic field effects on electrocatalytic reactions. Only nonmagnetic materials (i.e., Pt and Au) are used as electrodes, in order to rule out the influences caused by the magnetic properties of the electrode materials that might lead, for instance, to an enhancement in the reaction kinetics at ferromagnetic catalysts[3,7,17]. In addition, only the working electrode (WE) has been positioned in the magnetic field to avoid affecting the reference and counter electrode during the experiments. Electrocatalytic reactions involving gas-phase products, for instance, the OER and HER, are used to determine the magnetic field effects on the gas bubbles. In addition, the ORR, a reaction that consumes dissolved oxygen molecules and generates hydroxide ions ($OH^-$) in alkaline media, was used to study the magnetic field effects on the mass transport of ionic species without the influence of bubbles. We also used indicators, for example, $O_2$ bubbles from the chemical decomposition of hydrogen peroxide ($H_2O_2$) and pH-sensitive phenolphthalein dye ($In^{2-}$), to monitor the disturbed mass transport in the electrolyte under magnetic fields.

Using nonmagnetic electrodes, we introduced experimental protocols that allowed us to probe qualitatively and quantitatively the Lorentz force effect on modifying and, in some cases, enhancing mass transport. The results show that the major reason causing the change in mass transport in a magnetic field is the Lorentz force applied to diffusing ionic (charged) species. This force generates a whirling-like movement around the catalyst surface. In the case of gas-phase product reactions, the gas bubbles are good indicators of the Lorentz force. This Lorentz force effect plays a more predominant role in improving those reactions which have a low availability of reactants, for instance, the ORR, especially when the environment is not saturated with $O_2$.

## Results and discussion
### Origin of the magnetic field enhancement in mass transport
Our experiments were designed to decouple the effects on mass transport from the effects on the reaction kinetics. Our focus was to

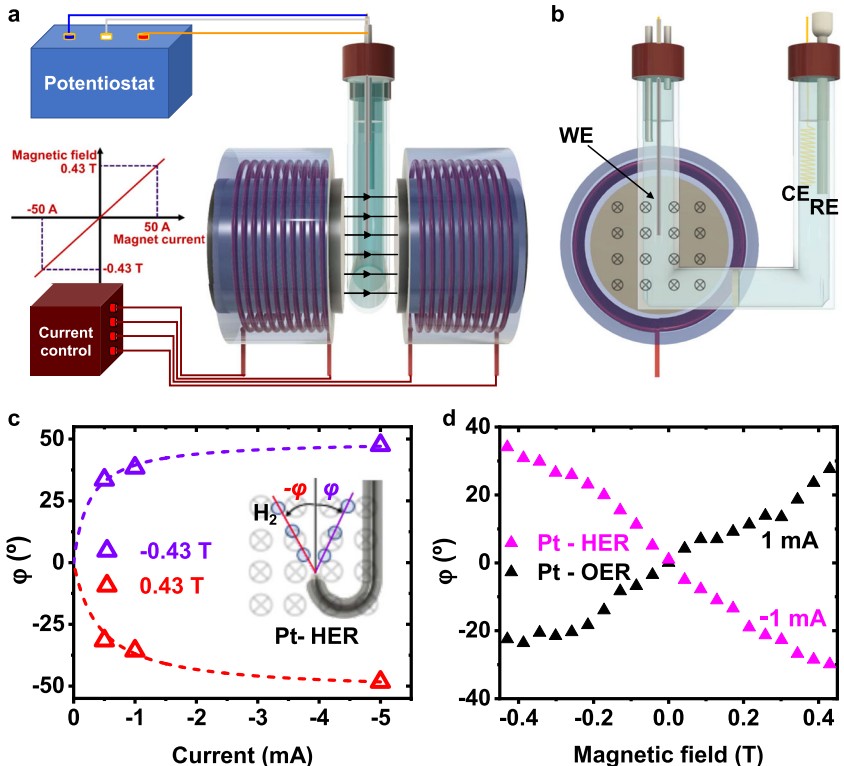

**Fig. 2 | Bubble stream movement in a magnetic field. a, b** Sketch of the magneto-electrochemical system and view of the electrochemical cell. **c** Variation of the angle of movement of the $H_2$ bubble stream with respect to the vertical ($\varphi$) as a function of the reaction current and the direction of the magnetic field. A negative magnetic field value refers to the same magnetic field strength but with the reverse direction. **d** Deviation angle ($\varphi$) for $H_2$ bubbles (HER) and for $O_2$ bubbles (OER), as a function of the magnetic field strength when the absolute reaction current was 1 mA.

reveal the main factor (e.g., the charged species) and determine the magnetic field-induced enhancement rate ($\eta = \frac{i_{field} - i_{no\,field}}{i_{no\,field}} * 100\%$) in different types of electrocatalytic reactions. Therefore, all catalysts used in this work were made of nonmagnetic pure Pt or Au.

We first used two gas bubble-evolving reactions, that is, the HER (Eq. (1)) and the OER (Eq. (2)) that take place in alkaline medium.

$$2H_2O + 2e^- \rightarrow H_2(g) + 2OH^- \tag{1}$$

$$4OH^- \rightarrow O_2(g) + 2H_2O + 4e^- \tag{2}$$

The magneto-electrochemical system is shown in Fig. 2a, b and Supplementary Fig. 1. The magnetic field only affected the working electrode (WE). To better determine the magnetic field effect on the movement of gas bubbles, we used an isolated Pt wire as WE, of which only the tip end was exposed to the electrolyte and therefore the gas evolution was restricted to that area. Without additional forces or disturbance, gas bubbles generated on the Pt surface move upward straight. When the magnetic field was turned on, the gas bubbles were pushed to one side depending on the direction of the magnetic field and the reaction current. In the case of the HER, the angle of the hydrogen bubble stream with respect to the vertical ($\varphi$) as a function of the current is shown in Fig. 2c. We clearly observed that a higher HER current led to stronger forces on the hydrogen bubbles, and that the bubble stream went in the opposite direction when the magnetic field was changed 180°. In Fig. 2d, the effect of the magnetic field on the bubble stream angle is plotted for both the HER and the OER with the same absolute reaction current. The angle is roughly proportional to the applied magnetic field.

Bubbles generated in the anodic (OER) and cathodic (HER) reactions were pushed in opposite directions (Fig. 2d). Videos and photos of the bubble movements are in Supplementary Movie 1 and Supplementary Fig. 2, respectively.

These experiments strongly confirm that an external magnetic field generates a force/disturbance that substantially influences mass transport. Its dependence on the reaction current and the magnetic field suggests that the changes are due to the Lorentz force. However, it is crucial to reveal the origin of this force/disturbance, especially whether the forces act directly or indirectly on the gas bubbles.

Other experiments with a Pt mesh show a whirling motion of the bubbles during the OER in the presence of the magnetic field (Supplementary Movie 2). We observed that changing the magnetic field direction with the electromagnet (Supplementary Movie 3) or locating a permanent magnet below the electrode (Supplementary Movie 4) lead to the bubbles curling in different directions. We noted that the force on the gas bubbles vanished immediately when the reaction was turned off, although the magnetic field remained (see the last seconds of Supplementary Movie 3). Therefore, we can conclude that the reaction current is essential to the generation of the forces on the bubbles.

The following possible hypotheses for the observed behavior are considered: either the freshly produced bubbles acquire charge and are influenced directly by the Lorentz force[11], the produced bubbles move upwards leading to the movement of ions in the surroundings, and these ions are deviated by the Lorentz force[14,15], or the moving ions in the solution are subject to the Lorentz force and push the bubbles. In order to test this, we carried out experiments using reactions that do not release gaseous products on the surface of the WE, for example, the ORR (Eq. (3)), in 0.1 M $HClO_4$ in the presence of 0.49 M $H_2O_2$.

$$O_2(aq) + 2H_2O + 4e^- \rightarrow 4OH^- \tag{3}$$

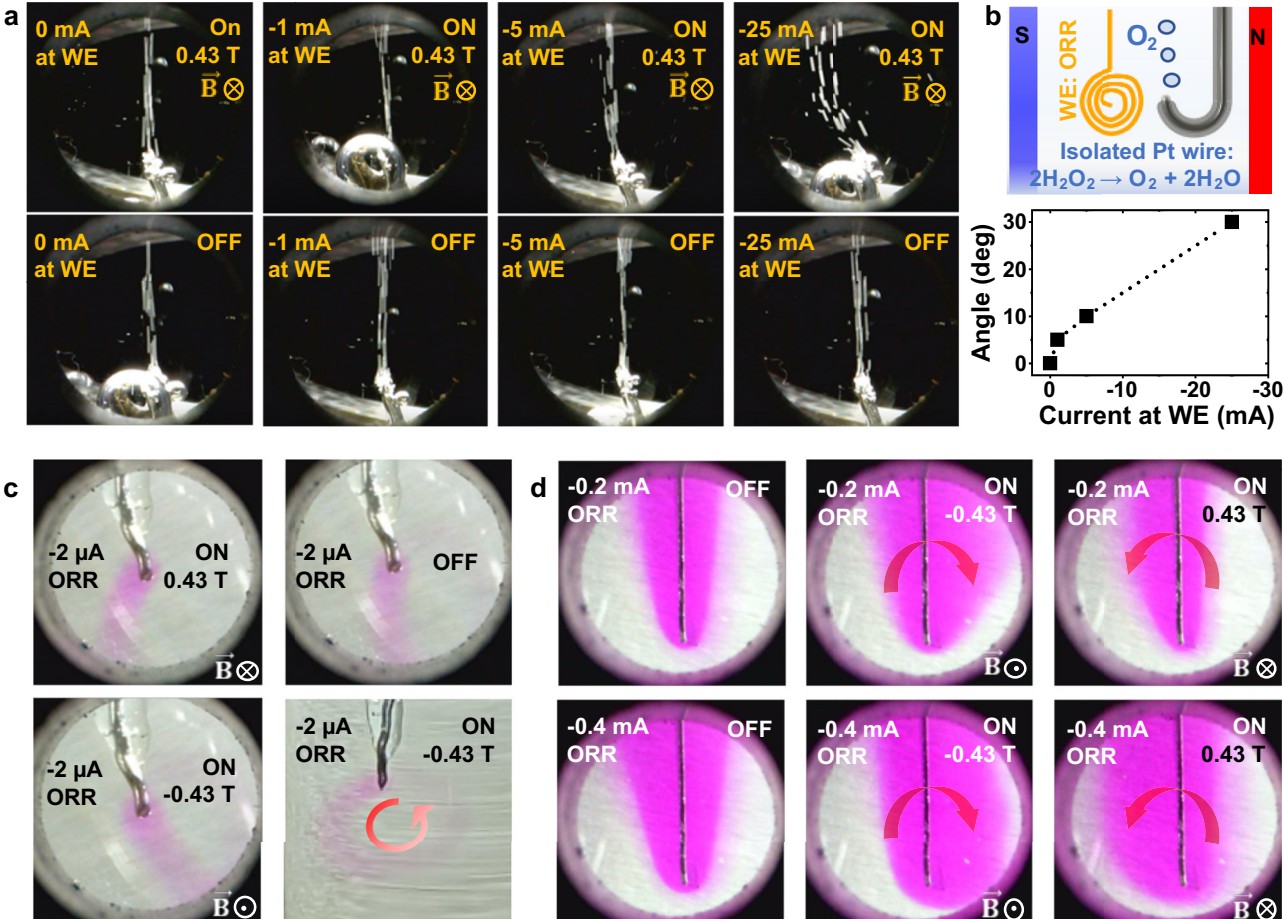

**Fig. 3 | Origin of the magnetic field enhancement in mass transport. a** Photos (taken through a hole in the magnet pole) of the movement of spontaneously generated O₂ bubbles at a Pt wire. A force in the electrolyte is active only when the ORR takes place at a nearby Au coil (Au not shown in the picture). **b** The schematic drawing of the system and the bubble deviation angle as a function of the ORR current at the Au WE. **c** Photos of solutions containing phenolphthalein pH indicator showing the concentrated OH⁻ region generated from an isolated Pt wire under different magnetic fields. The first three photos were taken through the hole in the magnet pole, and the fourth was taken from a side of the cell. **d** Photos (taken through a hole in the magnet pole) of a bare Pt foil during the ORR in the electrolyte containing the phenolphthalein dye. The arrow indicates the Lorentz-induced motion associated with the transverse diffusive current.

We changed the WE to a bare Au coil. An isolated Pt wire was placed next to the Au coil (Fig. 3a, b) to facilitate the spontaneous decomposition of H₂O₂ into O₂ bubbles and water (Eq. (4)) to monitor the disturbance in the electrolyte around the Au coil.

$$2 H_2O_2 \rightarrow O_2(g) + 2 H_2O \qquad (4)$$

The Pt wire was not connected to the potentiostat. Therefore, the electrochemical processes involving diffusion and migration of ionic species occurred only at the Au coil. Photos taken in different conditions are displayed in Fig. 3a. The O₂ bubble stream was not disturbed when no reaction took place at the Au coil, or when the magnetic field was turned off. We observed that the O₂ bubbles followed a vertical path, when the current at the Au coil was 0 (i.e., no electrochemical reaction is occurring), both in the presence or absence of the magnetic field (left-most panels of Fig. 3a). This suggests that even if charged particles were generated on the surface of the O₂ bubbles, the forces acting on these charges are not strong enough to affect the bubbles' behavior visibly. Additionally, the bubble stream was not disturbed in the absence of magnetic field, regardless of the current applied at the Au coil, indicating that neither the electric field—caused by the applied bias—nor the concentration gradient—caused by the electrochemical reaction—has an influence in the bubble stream direction. However, the bubbles started to move in lateral directions only when the ORR

occurred on the Au foil under a magnetic field. The movement angle and the ORR current followed a nearly linear relationship (Fig. 3b). Furthermore, a whirling motion of the bubbles was observed when the current at the Au coil was high (Supplementary Movie 5). These experiments demonstrate that the electrochemical reaction at the Au electrode generates ions which are then subject to the Lorentz force, thus stirring the electrolyte. Direct magnetic field effects on the formation of O₂ bubbles and their motion in the electrolyte were not noticeable.

To further visualize this magnetic field effect without using the movement of the bubble stream, we carried out ORR measurements at an isolated Pt wire (WE) in a solution containing phenolphthalein dye. Phenolphthalein is a chemical dye that turns purple in an alkaline environment, as shown in Eq. (5).

$$H_2In\,(colorless) + 2OH^- \rightleftharpoons In^{2-}\,(purple) + 2H_2O \qquad (5)$$

We added ten drops of phenolphthalein solution (1% solution in 85% ethanol) to a pH-neutral 0.1 M Na₂SO₄ electrolyte, which resulted in a colorless solution. Once the ORR started at the electrode, the solution around the Pt surface turned purple, as shown in Fig. 3c, implying an increase in pH due to the locally generated OH⁻. Without an external magnetic field, the OH⁻ ions stayed around the Pt surface and slowly moved toward the other side of the cell (which hosted the

RE and CE). The magnetic field caused the OH⁻ ions to move laterally in opposite directions depending on the direction of the field. A photo taken from the side of the cell depicts a ring-shaped high OH⁻ path caused by a rotational force (Fig. 3c).

When using a Pt foil as the WE, which was oriented so that the magnetic field was in-plane (Fig. 3d), the electrolyte around the foil turned purple after the ORR started at −0.2 mA. The purple area expanded as OH⁻ ions were continuously generated and diffused away from the foil. An asymmetry in the OH⁻ concentration was created by the magnetic field. Supplementary Movie 6a–c shows the movement of OH⁻ under a magnetic field of 0.43 T or in zero field. The asymmetry is even stronger when a higher current of −0.4 mA is applied, as shown in Supplementary Movie 7a–c. Furthermore, two additional observations could be made (from Supplementary Movie 7) when the reaction current was changed to +0.4 mA from −0.4 mA. First, the purple-colored region moved immediately towards the opposite side of the cell, indicating that the sign of the applied current highly influences the direction of movement of the ions. A rotatory movement was observed that was in conformity with the direction of the magnetic field and the ionic diffusive currents. Second, the purple color started disappearing near the electrode as a result of the consumption of OH⁻ by the OER (Eq. 2).

As discussed in previous studies[3,7,12,18,19], an external magnetic field-induced effect is potentially able to promote mass transport. The so-called magnetohydrodynamics (MHD) was introduced decades ago[20] to study of the behavior of electrically conducting fluids in magnetic fields and was successfully applied for plasma confinement in fusion reactors and liquid metal control[13]. The MHD is applicable to electrolytes[21–23]. However, several studies[3,24] have recently stated that the effect on mass transport is not visible or minor. Moreover, the MHD mechanism requires at least one charged object moving across a magnetic field. In the case of water electrolysis, the charged object that receives the magnetic field-induced forces was not clearly revealed.

The measurements using gas bubbles and pH indicators show the rotatory movement in the electrolyte around the electrode under a magnetic field. The movement is perpendicular to the magnetic field. Moreover, the gas bubble movement is a secondary effect. In Fig. 3a, the $O_2$ generated on the Pt wire started to move horizontally only once the ORR at the nearby Au coil was "turned on" with a magnetic field. The evidence gathered in the experiments presented above demonstrates clearly that a magnetic field-induced mechanical force is applied on the species involved in the ORR, HER and OER (for instance, OH⁻), but not on the $O_2$ bubbles directly.

Based on our results, we conclude that the magnetic field-induced mass transport enhancement is caused by the Lorentz force on the moving ionic (charged) species. The magnetic field has a stirring effect in the electrolyte around the electrode, which facilitates diffusion and convection. In contrast, the direct magnetic effect on the gas bubbles ($O_2$ and $H_2$) is negligible. The paths of the ions and gas bubbles typically have circular shapes because the Lorentz force follows the equation:

$$F_L = qE + qv \times B \tag{6}$$

where $F_L$ is the Lorentz force, q the charge, $E$ the electric field, $v$ the velocity of the charges and $B$ the magnetic field. In our case, the high concentration of electrolyte ensures $E$ ~0 except in the immediate vicinity of the electrode surface.

During electrochemical reactions, an ionic current emerges as a consequence of the applied bias and the depletion of reactants at each electrode. In the context of the OER in alkaline media, this ionic current primarily consists of anions migrating toward the electrode surface. Typically, it would mostly comprise OH⁻ ions as they are being consumed at the WE. These moving ions interact with the magnetic field, resulting in a change in their trajectory induced by the Lorentz force.

This force is perpendicular to both the ion's velocity and the magnetic field direction, as described in Eq. (6). Consequently, this interaction induces a rotational movement of the anions, deviating from their usual linear path toward the electrode when no magnetic field is applied. An illustrative representation of this phenomenon is shown in Fig. 4. In the case of the HER and ORR, the direction of the ionic current is reversed. In alkaline media, one can observe anions receding from the electrode surface (mainly OH⁻ in alkaline conditions) or cations approaching the surface (mainly H⁺ in acidic conditions). In this context, with the ionic current having an opposing polarity, the Lorentz force also reverses its direction, leading to the formation of a swirl in the opposite sense.

The experiments using the phenolphthalein dye allowed us to have a quantitative evaluation of the mass transport process. We have further analyzed luma (brightness) values from the video captures as a function of the distance from the electrode surface in Supplementary Movie 6a–c. The dye in an acidic environment is colorless, with a luma value of 255 due to the white background. The dye becomes purple in an alkaline environment, with a maximal luma value of 130 (saturated). The luma value acts therefore as an estimator for the concentration of reacted dye and an indication of the local OH⁻ concentration. Figure 5a shows the luma values at both sides of the electrode following similar curves over time when the magnetic field was off. At first, the image was mostly white, and the luma values were close to the maximum ($t = 0$). As the dye started getting colored by the OH⁻ produced on the electrode surface, the luma values started decreasing. The luma values on the left side of the WE are slightly lower at larger distances (>3 mm) due to some shadows in the background as the light was shone from the right side of the cell. The obtained curves are nearly symmetric around the electrode, as expected for an undisturbed mass transport process by migration and diffusion.

On the other hand, the curves obtained when the magnetic field was turned on are asymmetric, as shown in Fig. 5b, c. When the magnetic field is turned on to the positive direction (i.e., coming out of the plane of the picture, Fig. 5b), luma profiles on the right side of the WE extend farther away from the electrode surface than those from the left side of the WE. This indicates that OH⁻ ions reach farther out on the right side of the cell. In contrast, profile curves from the left side of the WE extended farther than profiles from the right side when the magnetic field was changed to the opposite direction (Fig. 5c). Therefore, a magnetic field introduces an asymmetric disturbance in the OH⁻ movement from the electrode surface via the Lorentz force. The disturbance (enhancement or depression) depends on the direction of the magnetic field relative to the electrode surface.

For a more detailed analysis, the luma values were converted to pH values and thus to the OH⁻ concentration (see "Methods" for details). It should be noted that because the phenolphthalein dye has a color change between pH 8.6 and 10.6, any OH⁻ concentration below $4.10^{-6}$ M or above $4.10^{-4}$ M cannot be distinguished. Figure 5d shows the concentration profiles obtained for the different magnetic field settings after 4 and 160 s of reaction, on the right side of the WE. After a certain time, the distribution of OH⁻ ions was visibly dependent on the magnetic field condition (on or off) and its direction. The concentration profiles at different times were fitted with Eq. (7), according to the solution of the Fick diffusion equations[25]:

$$C = C_0 \left( 1 - \left( \frac{x}{2\sqrt{D_{eff}\,t}} \right) \right) \tag{7}$$

where $C_O$ is the concentration on the surface of the electrode, and $D_{eff}$ is the effective diffusion coefficient, in which electrical and magnetic effects are included. Our aim is to make a semiquantitative comparison in the simplest framework, having observed that the data had a reasonable fit using a fickian diffusion profile. We term $D_{eff}$ as an

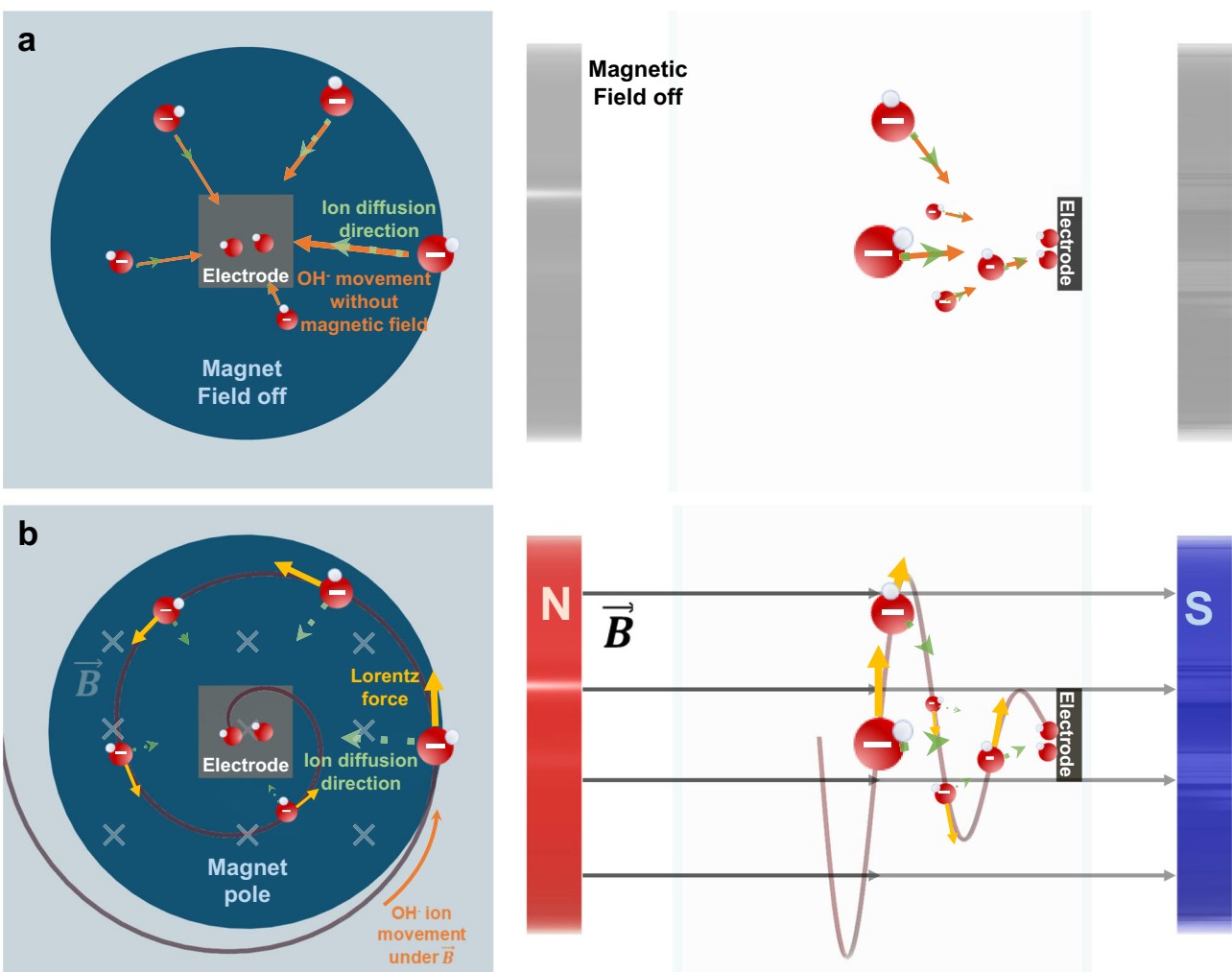

**Fig. 4 | Illustration of the Lorentz force on the diffusion of OH⁻ in the vicinity of the WE. a** During OER, OH⁻ ions diffuse towards the electrode where they are consumed in the reaction. **b** Under a magnetic field, the diffusion processes and the perpendicular Lorentz force result in a spiral movement of the OH⁻. Note that OH⁻ has a negative charge.

"*effective*" diffusion coefficient as the solution to Fick equations was obtained without considering the magnetic field or any kind of convection. Figure 5e shows an example of the curves for the situation with the magnetic field on (others can be seen in Supplementary Fig. 3). The effective diffusion coefficient was extracted by linear fit of the parameter $\lambda = 2\sqrt{D_{eff}t}$, as shown in Fig. 5f. Our experiments show that the magnetic field clearly induces an enhanced diffusion, equivalent to increasing the diffusion coefficient by approximately 45% toward one direction in the first minutes of the reaction, or decreasing it to one-third in the other. At longer times, the difference in the diffusivity of OH⁻ was even greater, as the diffusion length seemed to reach a plateau when the magnet was turned off, while it continued to increase when the magnetic field was on (shown in Supplementary Fig. 4). A similar trend was observed at the left side of the WE, although the estimation of $D_{eff}$ was not possible in this case given that the system was illuminated from the right side, and shadows from the WE hinders the accurate estimation of OH⁻ concentration (see Supplementary Fig. 5 for details).

**Utilization of the magnetic field enhancement in mass transport**
The effect of magnetic fields on mass transport is nowadays a matter of strong debate. Most studies have focused on the magnetic field effect on gas-phase product reactions. While ref. 3 states that this effect is not noticeable, others emphasized the importance of accelerated gas bubble removal caused by an external magnetic field[14–16,26–29]. However, most related studies focused only on gas bubble removal effects to improve the reaction rates. Such a Lorentz force effect in general electrocatalytic reactions has not been well explored, which potentially limits its wider utilization. Only a very recent study described the advantages of magnetic field-enhanced mass transport during electrochemical $CO_2$ reduction[19]. Thus, in the following we provide a more quantitative evaluation for OER and ORR that we believe will be useful for further process design.

We conducted the OER at Pt surfaces to evaluate the enhancement in the overall current due to mass transport effects induced by a magnetic field. The OER currents under different magnetic fields are shown in Fig. 6a. It is noticeable that the field enhanced the OER. The enhancement is dependent on the reaction current and the magnetic field. For instance, at 2.0 V, the enhancement in the OER current is -0.27 mA under 0.215 T and 0.41 mA under 0.43 T, but approximately 0.5 mA (0.215 T) and 0.85 mA (0.43 T) at 2.4 V. However, the enhancement rate (η) is relatively constant at different potentials under the same magnetic field, i.e., around 2.5% under 0.215 T and 4.0% under 0.43 T. Nevertheless, the magnetic enhancement in the OER current is much less compared to the values reported from magnetic

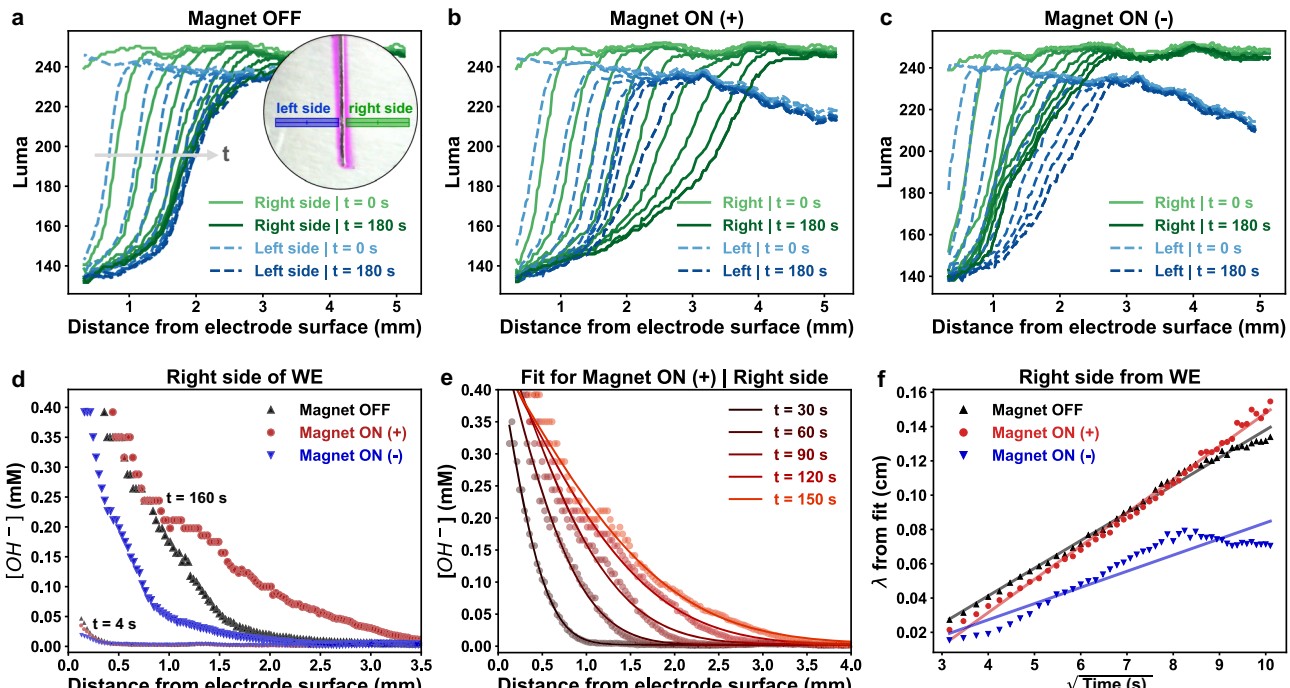

**Fig. 5 | Quantification of the mass transport effect. a–c** The phenolphthalein dye in its purple (basic) form produced a decrease in the luma values, which acts as an indicator of the diffusion of $OH^-$ ions. Luma profiles (related to color intensity, where white = 255, black = 0) are reported as a function of distance to the electrode at increasing times (*t*). The data were obtained from video analysis, when the magnetic field was turned off (**a**), turned on in one (**b**) and the other direction (**c**). Inset in a shows the line profiles where the luma was measured. **d** $OH^-$

concentration profiles at *t* = 0 and 180 s. **e** Fittings of the concentration profiles at different times with Eq. (7), where $\lambda = 2\sqrt{Dt}$ is the diffusion length. **f** $\lambda$ vs. the square root of time. The diffusion coefficient (*D*) for each condition was estimated from the slope of the plot: $D_{eff,OFF} = (6.5 \pm 0.1) \times 10^{-5}\ cm^2\ s^{-1}$, $D_{eff,ON(+)} = (9.4 \pm 0.1) \times 10^{-5}\ cm^2\ s^{-1}$ and $D_{eff,ON(-)} = (2.2 \pm 0.4) \times 10^{-5}\ cm^2\ s^{-1}$. These values compare well with the theoretical value of $5.3 \times 10^{-5}\ cm^2\ s^{-1}$ for $D_{eff,OFF}$[40].

field-induced kinetics enhancement at ferromagnetic catalysts[3,7,17]. Supplementary Fig. 6 shows the OER at a Pt microelectrode. The mass transport limitations are minimized at microelectrodes[30]. Therefore, a noticeable enhancement can only be seen at high overpotentials in Supplementary Fig. 6b.

When performing OER on model flat Pt foil surfaces, gas bubble removal is not an issue. We switched to a Pt mesh, a common support used in electrocatalytic studies, to determine how the magnetic field helps remove the gas bubbles. As can be seen in Supplementary Movies 3 and 4, the produced oxygen bubbles stuck in the Pt mesh regardless of the magnetic field. The rotational motion generated by the magnetic field was not able to sufficiently remove the gas bubbles in the mesh. As shown in Fig. 6b, a field of 0.43 T did not influence the OER current evidently. In Fig. 6c, the chronopotentiometry curves at different currents were recorded while switching the magnetic field. The reduction in the overpotential due to the magnetic field is marginal, i.e., nearly 0 mV at 25 and 50 mA and only 3.3 mV at 100 mA. Hence, it is reasonable that many previous studies did not recognize the magnetic field effects on mass transport.

It should be noted that mass transport in an electrocatalytic reaction involves not only the removal of products but also the approaching of reactants. In general, one of the major mass transport issues in the OER and the HER is the removal of produced gas bubbles[31]. Other reactions, for instance, the ORR, are more likely to be affected by the diffusion process of the reactants. We used a Pt wire to test the magnetic effect in both acidic and alkaline media. The effect of a magnetic field of 0.43 T on the ORR at a Pt wire in acidic (1 M HClO$_4$) and alkaline (1 M KOH) are shown in Fig. 6d, e, respectively. An increase in the limiting diffusion current under magnetic field is observed, that attributed this effect to magnetic field-induced convection[10,18,19,32–34].

Supplementary Fig. 7 displays the CVs recorded in $O_2$-saturated KOH while switching the magnetic field from 0.43 to 0 T (off). In both media, the enhancement in ORR current density was evident and similar in the diffusion-controlled regions, ca. 51% at 0.4 V vs. RHE. Furthermore, the magnetic field affected the ORR roughly to the same level at all potentials, as the polarization curves under 0.43 T divided by the current magnification at 0.4 V vs. RHE caused by the field (i.e., $I_{0.43\ T}/I_{0\ T}$) overlap with those recorded under 0 T (Fig. 6f). The CVs of the Pt wires in both media saturated with $N_2$ are shown in Supplementary Fig. 8. The surface areas were obtained from the hydrogen UPD regions[35].

Mass transport is also sensitive to lower magnetic fields. In the CVs shown in Supplementary Fig. 9, the increase in the ORR current at a Pt wire was found to be proportional to the magnetic field. For instance, a field of 0.215 T led to an increase of ca. 29.8 μA in the current during the anodic scan at 0.4 V vs. RHE, and doubling the field to 0.43 T led to approximately a twofold increase (i.e., ca. 64.7 μA).

To observe more clearly the role of the magnetic field in changing the $O_2$ availability near the WE, we performed ORR with a Pt foil sample placed at two different locations: at the liquid–air interface (top) vs. far from the interface. Figure 6g shows that at the "top" position, oxygen availability was higher according to the ORR polarization curves recorded under the magnet OFF condition. However, the magnetic field-induced increase in the ORR current was higher when the Pt foil was far from the liquid–air interface. This can be understood as follows: $O_2$ concentration near the Pt foil is determined by its dissolution into the electrolyte at the air-liquid interface, governed by Henry's law, and its diffusion towards the WE. When ORR begins, $O_2$ gets consumed, leaving the area around the WE depleted of $O_2$. At this point, the current is determined by the arrival of $O_2$ from the bulk of the solution. When the WE is located far away from the air-liquid interface,

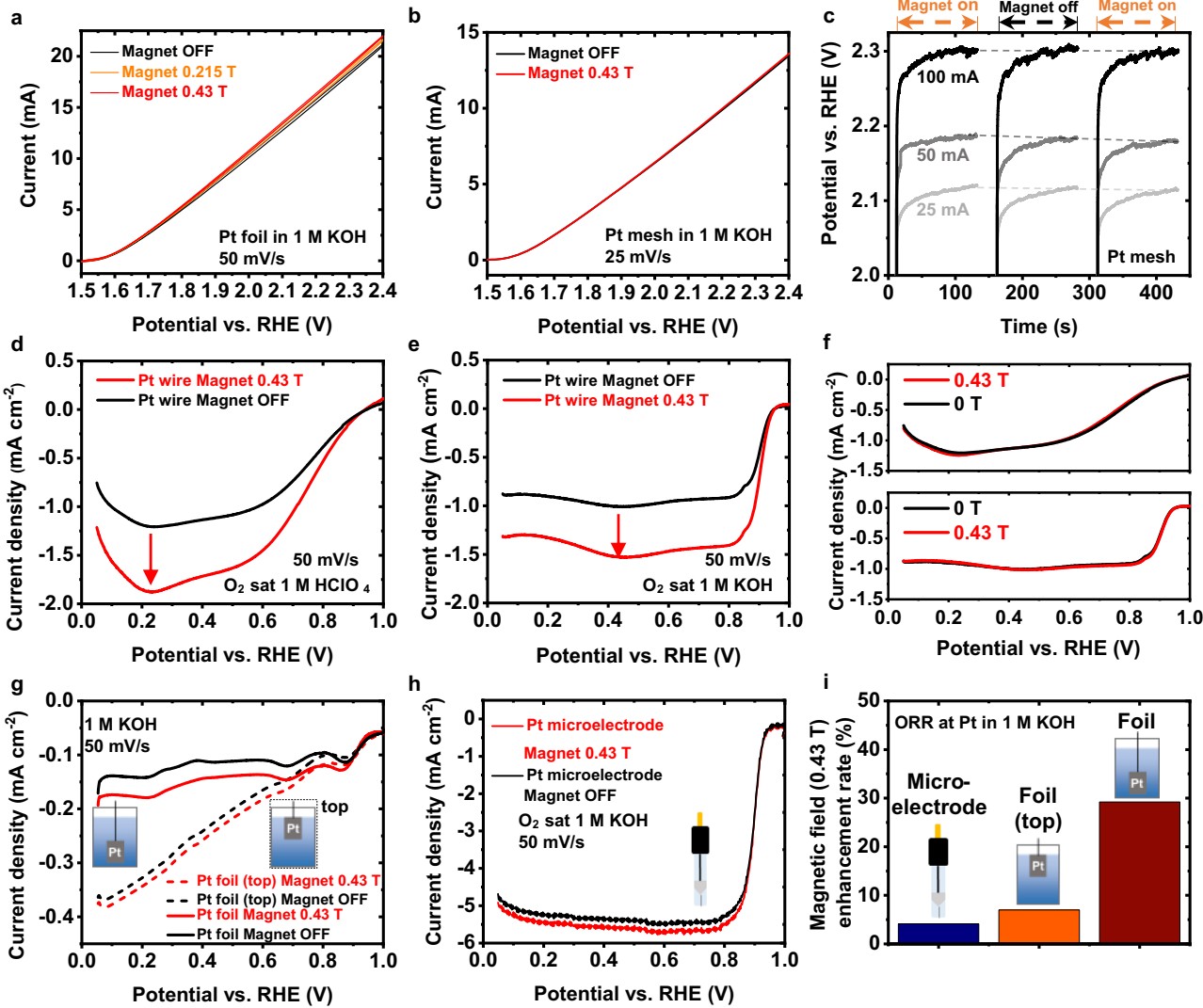

**Fig. 6 | Quantitative evaluation of the Lorentz force effect in electrocatalysis.** **a** The OER at a Pt foil in 1 M KOH under different magnetic fields. **b** The OER at a Pt mesh in 1 M KOH under different magnetic fields. **c** Chronopotentiometry measurements using a Pt mesh at different reaction currents. The magnetic field was turned from ON (0.43 T) to OFF and back to ON. Each period is 2 min. **d** ORR polarization curves at a Pt wire in acidic media. **e** ORR polarization curves at a Pt wire in alkaline media. **f** Normalized ORR polarization curves from (**d**, **e**). **g** Comparison of the enhancement in the ORR at a Pt foil when the foil was placed at different locations. **h** Magnetic field effect on the ORR at a Pt microelectrode. **i** Comparison of the enhancement rate at different Pt electrodes and locations. Analogous results were obtained with the magnetic field in the opposite direction (i.e., −0.43 T) in all cases.

the Lorentz force-induced stirring of the electrolyte facilitates the transport of dissolved oxygen from the air-liquid interface to the WE, making it more accessible for the ORR. Therefore, the magnetic field enhancement in the mass transport is more evident when the reactant concentration is low.

We also conducted the ORR at a Pt microelectrode (ø=10 μm). The ORR polarization curves are shown in Fig. 6h. The magnetic field did not influence the kinetics- and mixed-controlled regions noticeably. A small enhancement can be seen in the diffusion-controlled region. A comparison of the enhancement rate in the ORR at different Pt electrodes is shown in Fig. 6i. The enhancement rate was only around 4% in the case of Pt microelectrode. It increased to around 7% in the case of a Pt foil located at the liquid–air interface and nearly 30% when it was distant from the liquid–air interface.

To summarize, in the case of OER, where the reactant is typically the solvent (H₂O, or OH⁻ in alkaline media) and is constantly available near the WE, changes in the mass transport of the electrolyte caused by the external magnetic field do not significantly impact the reaction current. While the Lorentz force-induced stirring could assist in

removing the gas bubbles adhered to the electrode, our results indicate that the enhancement in the overall reaction current is small in the OER. The Lorentz force is not sufficient to accelerate the bubble removal. This explains why the effect of the magnetic field on mass transport is often overlooked in gas-phase product reactions. In contrast, this rotatory force helps the approach of a low-concentration reactant to the electrode surface, such as dissolved O₂ in ORR. For this reason, the magnetic field effect on the ORR is much more significant (up to 30% in the bottom position for the Pt foil, up to 51% for the Pt wire).

In previous works, the magnetic field was shown to promote HER and OER, but the reason behind this enhancement is not yet completely understood. In this work, we distinguish between kinetic and mass transport effects. We focus on the magnetic field effects on the mass transport in energy-related electrocatalytic reactions using nonmagnetic electrodes. We provide distinctive insights on the extent of the possibility of using magnetic fields for mass transfer enhancement and offer an explanation for its mechanism. Our results demonstrate that the magnetic field-induced Lorentz force on the

moving ionic species is essentially responsible for a rotatory motion in the electrolyte that facilitates mass transport, which can almost double the diffusion of ions toward one side of the electrode. Understanding the role of the mass transport effects is essential to the reproducibility of magnetically enhanced electrochemical reactions. These effects are universal in reactions involving the diffusion and migration of charged species. The forced gas bubble movement is a secondary effect in this process. This enhancement is minor in gas-phase product reactions when the reactant is abundant at the electrode/electrolyte interface, for instance, the OER, and it is much smaller than values reported from magnetic field-induced kinetics enhancement at ferromagnetic catalysts. However, this Lorentz force-induced ionic motion can play a crucial role in electrocatalytic reactions bearing low availability of reactants (or diffusion-limited), e.g., the ORR. By quantifying the effects involved in interfacial magnetic enhancement, we shed light into the factors that determine magnetic enhancement in water splitting or related reactions and provide a strategy that can be easily implemented to boost electrocatalytic reactions on different materials.

## Methods

### Materials
Pt wire (diameter: 0.5 mm, 99.95%), Pt mesh (45 mesh woven from 0.198 mm dia wire, 99.9%) and Au wire (diameter: 0.813 mm, 99.9%) were purchased from Alfa Aesar. Pt foil was purchased from Vega y Camji (Argentina). KOH (1 N solution in water) was purchased from Acros Organics™ and used without further purification. Phenolphthalein dye (1% in ca. 85% ethanol, for titration) was from Chemie Brunschwig AG. Pt microelectrodes (diameter: 10 μm) were from BASi Research Products. 30% (w/w) aqueous $H_2O_2$ solution (puriss. p.a.) was from Sigma-Aldrich.

### Electrochemistry
Electrochemical experiments were performed in a borosilicate H-cell with either a Bio-Logic SP-150 or Autolab potentiostat, HydroFlex® standard hydrogen reference electrode (Gaskatel GmbH) and a Pt mesh counter electrode. The compartment containing the working electrode was placed between the electromagnet poles, i.e., inside the magnetic field. RE and CE were placed in the other compartment and outside the magnetic field, as shown in Fig. 1b. Except when otherwise noted, the electrolyte was 1 M KOH. Oxygen was purged into the electrochemical cell during ORR measurements. During microelectrode measurements, a homemade Faradaic cage covered the cell to reduce noise.

When microelectrodes were used as WE, they were made of Pt. A Pt foil, Pt and Au wires and Pt mesh were used as macro-sized WEs. The microelectrodes were polished and cleaned with Milli-Q (Millipore) water before each set of measurements. The Pt and Au electrodes were cleaned by flame annealing and rinsed with Milli-Q water. Pt foil and mesh were positioned parallel to the magnetic field lines (as shown in Supplementary Fig. 1 and Fig. 3d).

### Magnetic fields
The magnetic field was generated with an electromagnet (Lake Shore Cryotronics, Inc.). A Hall probe (Lake Shore Cryotronics, Inc.) was used to assess the magnetic field intensity at the position of the working electrode. Magnetic fields in the range of 0 to 0.55 T were used. The needed electromagnet current for each applied magnetic field was calibrated at the position of the WE with a Hall probe before each set of experiments. The uniformity around this area was verified with a Hall probe. The magnet poles ensure a uniform magnetic field between the two poles. The WE was carefully placed in the center of the poles. As the cell dimension (<3 cm) is much smaller than the pole diameter (10 cm), we can guarantee the magnetic field is uniform in the vicinity of the WE (which is nonmagnetic), as any inhomogeneities caused by border effects would be several cm away. The electromagnet is water-cooled, ensuring that no heat reaches the EC-cell. The field homogeneity is better than 1 mT, and its value is stable over this field range for days, while running an electrochemical experiment[36]. All experiments were done at room temperature. A photograph and a more detailed scheme of the setup is shown in Supplementary Fig. 1. A hole in one of the electromagnet poles allows for video recording with a digital microscope camera and to take images of the WE vicinity in the plane perpendicular to the magnetic field. To facilitate the visibility of the bubbles or the pH indicator, a black or white piece of paper, respectively, was located behind the EC-cell as background.

### Bubble stream angle determination
For the studies evaluating the bubble stream formed on the electrode by HER or ORR (Fig. 2b, c and Supplementary Fig. 2), a Pt wire was folded into a J shape and covered with hot glue (Pattex, based substance of preparation: ethylene-vinyl acetate copolymer and hydrocarbon resins) to isolate the surface except for the very end of the wire. This hot glue is typically used for electrochemical scanning tunneling microscope tip isolation. The angles formed between the normal to the electrode surface and the stream of bubbles traveling to the top were measured with *Image J* software[37] from a set of images obtained at various magnetic fields and reaction currents. The same isolated Pt wire was used for spontaneous $H_2O_2$ decomposition. In this case, a nearby Au coil was connected to the potentiostat instead of the Pt wire (Fig. 3a, b). The potential and current at the Au coil were controlled as shown in Supplementary Fig. 10.

### Measurements with a pH indicator
ORR measurements were conducted on a Pt foil parallel to the magnetic field. The electrolyte was 0.1 M $Na_2SO_4$ with 10 drops of phenolphthalein 1% solution in 85% ethanol as the pH indicator. Videos were recorded through a hole in the electromagnet pole during chronopotentiometry control at different electrode currents (−0.2, −0.4, and +0.4 mA). A typical constant current control and the resulting potential plot are shown in Supplementary Fig. 11). The magnetic field was set to 0, −0.43, and +0.43 T, respectively.

### Quantitative video analysis
The videos recorded at −0.2 mA were analyzed with the software *Tracker*[38]. The distance in the video size was calibrated using the Pt foil thickness of 0.4 mm. Two linear profiles were drawn on each side of the electrode for luma tracking. The luma value represents the brightness of a spot in a video. Position and luma data were extracted every 2 s for 3 min, taking $t = 0$ right before the appearance of the pink color. The absorption of the phenolphthalein dye in the pink region is known to follow a sigmoidal curve around its pKa ~9.5. In its acid form, the dye is transparent when the dissociation degree ($\alpha_{In^-}$) is close to 0. Its basic form is pink, with a maximum of absorbance at 523 nm. Absorbance (A) data as a function of pH was extracted from ref. 39 and converted to luma values assuming that A = 0 ($\alpha_{In^-} = 0$) corresponds to luma = 250 (maximum value in the analyzed videos), while A = 0.375 ($\alpha_{In^-} = 1$) corresponds to luma = 130. The extracted data were then fitted with a sigmoidal (logistic) curve. The result of this fit was used to convert the luma values to local pH, which was further transformed into $OH^-$ concentrations. The resulting concentration profiles ([$OH^-$] vs. distance) were fitted with Eq. (7), which describes the diffusion profiles of a species assuming a constant concentration $C_0$ at $x = 0$. In our case, $C_0$ is given by the $OH^-$ generated on the surface of the electrode, which should be constant as the applied current was maintained at −0.2 mA during the reaction. Finally, the effective diffusion coefficient ($D_{eff}$) was estimated from the slope of a linear fit of $\lambda$ vs. the square root of time. A similar analysis was done by estimating the $OH^-$ diffusion length ($\lambda$) by measuring the position where the same luma value is obtained at each time and estimating $D_{eff}$ from $\lambda = 2\sqrt{Dt}$, yielding the same results (Supplementary Fig. 12).

## Data availability

The data generated in this study are provided in the Supplementary Information and the original dataset is available in the Source Data file. Source data are provided with this paper.

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

## Acknowledgements

P.V. and Y.L. thank Mika Tamski and Jonas Milani for their assistance with the electromagnet setup. P.V. thanks the Swiss Federal Commission for Scholarships for Foreign Students (FCS) for the Swiss Government Excellence Scholarship. P.V. and G.J.A.A.S.I. thank the financial support of CONICET. P.V., Y.L. and M.L. acknowledge the financial support from the European Union's Horizon 2020 research and innovation programme (732840-A-LEAF) and the Max Planck-EPFL Center for Molecular Nanoscience and Technology.

## Author contributions

P.V. and Y.L. designed and performed the experiments and analyzed the data. M.L. designed the research plan, research grants, and supervised the project. J.-P.A. participated in the experimental design. G.S.-I. participated in the data discussions. All authors participated in the data interpretation. The manuscript was written by P.V. and Y.L. with contributions from all the authors.

## Competing interests

The authors declare no competing interests.
