## [Peer Review File · Nature Communications]

Reviewers' comments:

Reviewer #1 (Remarks to the Author):

Recently, a magneto-electrochemical effect has attracted much attention in electrochemistry related fields of electroplating, corrosion, batteries and so on, because magnetic fields can affect the kinetics and mass transfer behaviors in the electrochemical reaction process through various mechanical forces such as Lorentz force, magnetic field-gradient force (Kelvin force), paramagnetic force, and damping force. Therefore, understanding and controlling the magneto-electrochemical effect and its behavior is of great scientific and application value for the development of electrochemistry related research fields and industries. Although the development of magneto-electrochemistry is inseparable from the guidance of the fundamental theories of physics, there is still no clear explanation of the specific object of action and the influence mechanism of the magnetic field on the electrochemical reaction process. At present, most of the magneto-electrochemistry related research is only carried out for practical application, and the in-depth research on the influence mechanism is still very lacking.

In this manuscript, the experimental conclusion that the charged gas bubble products (e.g., H₂ & O₂) are subject to the Lorentz force of the magnetic field [7] and the simple conclusion that the magnetic field affects the electrochemical reaction [3] are questioned, and ingenious experiments are designed to confirm the imperfections of the above conclusions by visual qualitative methods and approximate quantitative methods. This practice of questioning and testing existing research conclusions is to be highly encouraged. However, if the manuscript is to be published in NC, the current version still lacks some indisputable evidence support, such as:

1) During the decomposition of H₂O₂, how to ensure that no charged particles are generated on the surface of O₂ bubbles? If a thin tube is used to blow bubbles into acidic, alkaline, and neutral deionized aqueous solutions, will the same conclusion still be obtained under the action of external magnetic field?

2) In this manuscript, the explanation of the phenomenon in Figure 6g-6i is relatively simple. According to the enhancement effect of the magnetic field on the electrochemical reaction in the three cases, it is hastily attributed to the difference of availability O₂ in the ORR process and the OER case on the Pt foil. I believe that the difference in the availability of O₂ for ORR should be related to its reaction kinetics. In the case of Pt microelectrodes, the adsorption effect of surface gas products on the surface of the microelectrode may be involved. Please explain in detail.

3) In the experimental videos and images, some experimental backgrounds are black (Fig. 3a) and some are white (Figs. 3c and 3d), why is this?

4) In this manuscript, an electromagnet is used to provide an external magnetic field for experiments. In order to exclude the factors caused by magnetic field gradient, is the uniformity of the magnetic field distribution in the effective region (WE region) of the electrochemical reaction evaluated?

5) In this manuscript, only the relationship between OH⁻ diffusion length (λ) and the square root of time was presented on the right side of WE, but the situation on the left side of WE is not presented. For better comparison, please also show the situation on the left side of WE.

Reviewer #2 (Remarks to the Author):

In this study, Vensaus et al. explored the impact of magnetic fields on mass transport within a specially designed magneto-electrochemical system. Their findings revealed a ~7% enhancement in the limiting current of the oxygen reduction reaction (ORR) on a Pt foil, while no significant effect was observed for the oxygen evolution reaction (OER).

Nevertheless, this slight improvement in the ORR limiting current through the application of a magnetic field has no practical significance, considering the possibility of other more cost-effective approaches to achieve a similar outcome. Additionally, the study suffers from methodological issues, including the adoption of inappropriate techniques and the omission of several factors that can influence the limiting current of ORR. These limitations raise doubt on the validity of the conclusions drawn by the authors. Furthermore, the so-called magnetohydrodynamics was introduced decades ago. While similar investigations into the influence of magnetic fields on mass transport have been conducted in related fields. For instance, a recent publication 'Magnetic Fields Enhance Mass Transport During Electrocatalytic Reduction of CO₂' (Chem Catalysis 2, 797–815, 2022) has provided analogous findings.

Given the lack of novelty in the current study and its limited practical significance, I am unable to recommend its publication in a prestigious journal like Nature Communications.

Other important issues:

1. A critical issue arises from the methodology employed: all the conclusions drawn by the authors lie on the utilization of phenolphthalein dye (H₂In, colorless) as a chemical indicator. This dye undergoes a color change to purple (In²⁻, purple) in an alkaline when it interacts with OH⁻ ions. However, it's

important to note that once phenolphthalein dye forms a complex with OH⁻, the resulting purple species (In²⁻) also carry a negative charge. Consequently, any nonparallel movement of these charged dye molecules, possibly due to concentration gradients, would subject them to the influence of the Lorentz force under a magnetic field. This implies that all the observed behaviors analyzed by the authors pertain to the charged dye In²⁻, rather than the OH⁻ ions themselves. Furthermore, it is worth noting that some reports have suggested that OH⁻ ions may not physically migrate during electrochemical processes (e.g., proton hopping mechanism for OH⁻ ions “transfer”).

2. A lack of essential information regarding the applied magnetic field. Firstly, the direction of the magnetic field is not provided, as commonly understood, the Lorentz force acting on charged particles is direction dependent. The authors mentioned that "ORR measurements were conducted on a Pt foil parallel to the magnetic field," but there are two directions for parallel and they did not clarify the conditions for OER. Secondly, information of uniformity, strength gradient, distance between magnetic poles, and operating temperature is missing. Given that ORR is highly temperature-sensitive, the absence of details regarding the magnetic poles' temperature and the distance to the electrochemical cell raises concerns about the validity of the results.

3. The authors claimed that “a magnetic field-induced mechanical force is applied on the species involved in the ORR, HER and OER (for instance, OH⁻), but not on the O₂ bubbles directly.” (Line 244-246). However, in the case of ORR, the reactants in the electrolyte are not charged (O₂, H₂O), but the charged OH⁻ ions is the resultant. This raises a critical question: is the observed increase in limiting current a result of the magnetic field enhancing the diffusion of resultant OH⁻ ions? What is the underlying reason for this? Although the ORR does not directly produce gaseous products, it does require oxygen gas (O₂) as a reactant. Given that the Lorentz force affects charged ions in the electrolyte, whether this force generates any secondary effects that indirectly influence the behavior of oxygen bubbles, and consequently the ORR process?

4. The authors have not provided a clear explanation of the differential effects of mass transfer on the OER and ORR. The OER involves charged OH⁻ ions and produces O₂ gas, and the ORR involves O₂ gas and produces OH⁻ ions, but both OH⁻ and O₂ coexist in their electrolytes. If the Lorentz force induced by the magnetic field primarily acts on OH⁻ ions, but no improvement in the OER while an improvement in the ORR is observed, it raises questions that mass transfer of reaction products is important than that of reactants? Besides, as stated by authors that a secondary effect is imposed on O₂ bubble, but what is the difference for OER and ORR? Thus, further clarification about the effects on OH⁻ ions and O₂ bubbles for OER and ORR are needed.

5. The reliability of the observed increase in the ORR limiting current using this system is questionable. It is suggested to employ a standard Rotating Ring-Disk Electrode (RRDE) setup to enhance the credibility and consistency of the ORR tests. The RRDE configuration is widely accepted in electrochemical research

for ORR testing as it allows for the separation of reaction products (such as H₂O₂), quantitative analysis, and reliable comparisons.

6. Pt is not suitable as a working electrode (WE) for OER testing as it can undergo oxidation on the anode, leading to the dissolution of Pt ions into the electrolyte solution. The charged Pt ions with a concentration gradient can also be influenced by magnetic field. Thus, the designed system with Pt as WE not only contaminate the electrolyte but also affect the accuracy and reliability of the results.

7. The authors presented electrode currents, such as -0.2, -0.4, and +0.4 mA, without additional details. Raw currents lack meaningful interpretation, as they depend not only on the applied potential but also on the electrode area. It is suggested to normalize the currents by the electrochemical surface area (ECSA) or electrode area to enhance the clarity and relevance of these results. Additionally, it would be better to specify the techniques employed to measure these currents—whether CA or CP methods were used? The applied potential for each of these cases should also be explicitly stated to provide a comprehensive understanding of the experimental conditions.

8. It's important to clarify the basis for the depiction of the Lorentz force acting on the diffusion of OH⁻ ions in Figure 4. Is it derived from experimental observations or theoretical modelling? Firstly, as previously mentioned in point 1, there is no direct experimental evidence of OH⁻ diffusion presented in the study. Secondly, the work lacks the inclusion of theoretical models that could generate these results.

9. As per the authors' exposition, the flat Pt foil and Pt mesh manifest notably disparate OER behaviors, as evidenced in Figure 6a and b. The onset potential for the flat Pt foil is significantly diminished in comparison to the Pt mesh. It would be beneficial to elucidate the underlying reasons for this disparity. Additionally, a comprehensive explanation regarding how the geometrical configuration of the electrodes modulates mass transfer in the presence of a magnetic field is desirable.

10. Referencing Supplementary Figure 7, it's evident that magnetic fields exert a tangible effect on adsorptive behaviors exhibited on electrode surfaces. This could potentially correlate with alterations in the bilayer structure of the electrode surface, as underscored by prior works (J. Phys. Chem. B, 2001, 105, 9487–9502; J. Phys. Chem. B, 2004, 108, 5778–5784).

Reviewer #3 (Remarks to the Author):

Authors investigated the effect of magnetic field on mass transfer by means of optical vision and electrochemical testings. Authors have made some progress compared to the reported references, more

clearly demonstrated mass transport in the magnetic field. However, the innovation of the article needs to be discussed. Some comments are as follows,

1. What are highlights in this manuscript, which innovations make differences from other references such as Scientific reports 6 (1), 21068 (2016); Sci Rep 11, 9346 (2021);ACS Applied Energy Materials 3 (7), 6752-6757 (2020).
2. The mechanism of mass transfer is not clear, please adding simulation calculations.

Reviewers' comments:

Reviewer #1 (Remarks to the Author):

Recently, a magneto-electrochemical effect has attracted much attention in electrochemistry related fields of electroplating, corrosion, batteries and so on, because magnetic fields can affect the kinetics and mass transfer behaviours in the electrochemical reaction process through various mechanical forces such as Lorentz force, magnetic field-gradient force (Kelvin force), paramagnetic force, and damping force. Therefore, understanding and controlling the magneto-electrochemical effect and its behaviour is of great scientific and application value for the development of electrochemical related research fields and industries. Although the development of magneto-electrochemistry is inseparable from the guidance of the fundamental theories of physics, there is still no clear explanation of the specific object of action and the influence mechanism of the magnetic field on the electrochemical reaction process. At present, most of the magneto-electrochemistry related research is only carried out for practical application, and the in-depth research on the influence mechanism is still very lacking.

In this manuscript, the experimental conclusion that the charged gas bubble products (e.g., H₂ & O₂) are subject to the Lorentz force of the magnetic field [7] and the simple conclusion that the magnetic field affects the electrochemical reaction [3] are questioned, and ingenious experiments are designed to confirm the imperfections of the above conclusions by visual qualitative methods and approximate quantitative methods. This practice of questioning and testing existing research conclusions is to be highly encouraged. However, if the manuscript is to be published in NC, the current version still lacks some indisputable evidence support, such as:

1) During the decomposition of H₂O₂, how to ensure that no charged particles are generated on the surface of O₂ bubbles? If a thin tube is used to blow bubbles into acidic, alkaline, and neutral deionized aqueous solutions, will the same conclusion still be obtained under the action of external magnetic field?

We highly appreciate the reviewer's acknowledgment of the importance of our work. As the reviewer mentioned, our focus is on the mechanism of the Lorentz force from the magnetic field. Therefore, all the experiments were specially designed and conducted to this end, and we thank the reviewer for this suggestion.

While we did not explore the possibility of blowing bubbles into different environments, the experiment shown in Fig. 3a is similar to the bubble blowing suggested by the reviewer. The decomposition of H₂O₂ at the Pt did not involve considerable ion diffusion. The bubble movement is governed by the magnetic field and the ORR at the Au WE. Our observations consistently indicate that the magnetic field has no significant effect when the current at the working electrode is 0 mA (no applied potential). This is shown in Fig. 3a (left panels): when the current at the WE is 0 mA, the O₂ bubbles evolve following a vertical path, both in the presence (top) or absence (bottom) of the magnetic field. This suggests that even if charged particles were generated on the surface of the O₂ bubbles, the forces acting on these charges are not strong enough to affect the bubbles' behavior visibly. Thus, we would expect to obtain the same conclusion by using bubbles blown into the electrolyte. We anticipate that the bubble movement does not depend on the pH of the solution, but

on the charge of the dominant diffusing ions, under the same magnetic field. We appreciate the feedback and hope this explanation clarifies the basis of our observations. We added this explanation on page 8 of the manuscript.

2) In this manuscript, the explanation of the phenomenon in Figure 6g-6i is relatively simple. According to the enhancement effect of the magnetic field on the electrochemical reaction in the three cases, it is hastily attributed to the difference of availability O₂ in the ORR process and the OER case on the Pt foil. I believe that the difference in the availability of O₂ for ORR should be related to its reaction kinetics. In the case of Pt microelectrodes, the adsorption effect of surface gas products on the surface of the microelectrode may be involved. Please explain in detail.

We appreciate the thoughtful feedback from the reviewer. We would like to highlight that the results shown in Figure 6g-i are all related to ORR, and no gas products are involved.

We understand the concern of the reviewer that we attribute the enhancement only to magnetic field effects on mass transport. However, this is the focus of this manuscript, and we provide a more comprehensive explanation below. We also improved our description in the main text (pages 20-21).

The scenarios presented in Figure 6g-i were carefully designed to emphasize the role of O₂ availability near the WE during ORR, which is directly influenced by the mass transport within the electrolyte. In these experiments, we have chosen non-magnetic Pt electrodes, and the key factor at play is the availability of O₂. In the case of the Pt microelectrode (Fig. 6h), mass transport, and specifically O₂ availability, is not a determining factor. This is because the amount of dissolved O₂ around the Pt microelectrode is not affected by the ORR significantly, due to the micrometer-scale size of the electrode and the O₂-saturated environment. However, in the case of the Pt foil in air -without the purging of pure O₂ gas- (Fig. 6g), the depletion of O₂ around the electrode becomes a critical factor, making the magnetic field effect significant. As shown in Fig. 6i, when O₂ is highly available near the WE (either by means of using a microelectrode, where O₂ depletion is not significant, or by locating the Pt foil near the electrolyte/air interface, where dissolved O₂ is more readily available), the effect of the magnetic field is smaller. However, when O₂ availability is reduced (locating the WE further away from the electrolyte/air interface) the mixing generated by the magnetic field becomes more significant, thus leading to a larger enhancement.

We acknowledge the strong correlation between O₂ (or other reaction intermediates such as OH-) coverage and ORR kinetics. However, it is important to note that the scope of this manuscript is not centred on examining reaction kinetics. As the reviewer correctly pointed out, our experiments were carefully designed to focus specifically on mass transport effects. In the case of O₂-saturated experiments (Fig. 6 d, e, h), we took steps to ensure O₂ saturation by purging O₂ into the electrolyte for a minimum of 10 minutes before initiating electrochemical measurements. We also maintained O₂ saturation in the electrolyte by positioning the O₂ inlet above the electrolyte surface without creating O₂ gas bubbles in the electrolyte. Since the O₂ concentration in the electrolyte in these cases was rather constant, we assume the reaction kinetics were not affected by the magnetic field considerably. Therefore, we mainly looked at the limiting current of the ORR curves, as mentioned above. In the experiments with O₂-saturated electrolyte and Pt wire as WE (Fig. 6d-e), we also observed the effect of the Lorentz force-induced stirring, likely due to the depletion of dissolved O₂ near the Pt wire surface.

In the case of OER (Fig. 6 a-b) the effect of the magnetic field in the mass transport is less significant for the overall current, given that the reactant is the solvent (H_2O or OH^-), which is highly available at all times near the WE. The primary impact of the Lorentz force-induced stirring would be on the removal of O_2 bubbles adhering to the electrode, although our results show that this effect is not very significant.

We hope this clarifies the basis for our approach and our explanation of the magnetic field effect, and we appreciate the reviewer's valuable input.

3) In the experimental videos and images, some experimental backgrounds are black (Fig. 3a) and some are white (Figs. 3c and 3d), why is this?

We used a black background to facilitate the observation of bubbles, and a white background to observe the color change of the indicator. The background was obtained by placing a piece of paper (black or white) behind the EC cell. We thank the reviewer for the question, and we have clarified this in the Methods section.

4) In this manuscript, an electromagnet is used to provide an external magnetic field for experiments. In order to exclude the factors caused by magnetic field gradient, is the uniformity of the magnetic field distribution in the effective region (WE region) of the electrochemical reaction evaluated?

We carefully placed the EC-cell in a way such that the WE is located in the middle of the electromagnet poles. As the WE is much smaller than the poles, we can be sure that the magnetic field is homogeneous in this area. Any inhomogeneities caused by border effects would be at least 5 cm away from the WE. To our knowledge, this is the most accurate way to ensure a uniform field. A photograph of the setup and a more detailed scheme were added to the SI (Suppl. Figure 1). We calibrated the electromagnet current to the observed magnetic field measured in the position of the WE, and we verified the uniformity with a Hall probe. Refer to Blumenschein et al. (*Phys. Chem. Chem. Phys.* 2020, 22, 997) for evidence of the magnetic field homogeneity and stability in another electrochemical experiment conducted using the same magnet as the one used in this study. EPR spectra of 1 mT in width were recorded in a special data-taking scheme required for that other experiment. Thus, data points were accumulated over a time span of several days. Thus the field homogeneity is better than 1 mT and its value is stable over this field range for days, while running an electrochemical experiment.

5) In this manuscript, only the relationship between OH^- diffusion length (λ) and the square root of time was presented on the right side of WE, but the situation on the left side of WE is not presented. For better comparison, please also show the situation on the left side of WE.

We chose to show the results from the right side of the WE because the color profiles are clearer due to the light coming from this side. The left side has some shadows from the WE itself that lead to less clear profiles and a more difficult analysis. In any case, the situation on the left side was added to the SI.

Reviewer #2 (Remarks to the Author):

In this study, Vensaus et al. explored the impact of magnetic fields on mass transport within a specially designed magneto-electrochemical system. Their findings revealed a ~7% enhancement in the limiting current of the oxygen reduction reaction (ORR) on a Pt foil, while no significant effect was observed for the oxygen evolution reaction (OER).

Nevertheless, this slight improvement in the ORR limiting current through the application of a magnetic field has no practical significance, considering the possibility of other more cost-effective approaches to achieve a similar outcome. Additionally, the study suffers from methodological issues, including the adoption of inappropriate techniques and the omission of several factors that can influence the limiting current of ORR. These limitations raise doubt on the validity of the conclusions drawn by the authors. Furthermore, the so-called magnetohydrodynamics was introduced decades ago. While similar investigations into the influence of magnetic fields on mass transport have been conducted in related fields. For instance, a recent publication 'Magnetic Fields Enhance Mass Transport During Electrocatalytic Reduction of CO₂' (Chem Catalysis 2, 797–815, 2022) has provided analogous findings.

Given the lack of novelty in the current study and its limited practical significance, I am unable to recommend its publication in a prestigious journal like Nature Communications.

We thank the reviewer for the concerns. We would like to emphasize again that this manuscript is focused on the real mechanism of the magnetic field enhancement on mass transport of electrocatalysis. For the first time to the best of our knowledge, we accredited the enhancement to the Lorentz force acting on moving ions. Moreover, the Lorentz force acting directly on gas bubbles being the reason for mass transport enhancement has been disapproved.

Please note that 7 % is the minimum enhancement that we report for ORR, while 50 % was the largest observed. ORR is a well-known sluggish reaction in a number of important systems, including fuel cells. Enhancement of ORR, especially using an unconventional approach, is very valuable. Knowing the source of the enhancement we can find the conditions to improve upon this even further. Also, we believe that the application of a magnetic field could be combined with other methods to improve ORR, and does not significantly add to the cost as it could be done with a permanent magnet. In our case we used an electromagnet for a detailed controlled study, but this would not be necessary in practical applications.

While we agree that magnetohydrodynamics was introduced decades ago, we observed that the effect is not clearly determined in electrocatalytic systems. Literature searching indicates that different hypotheses explain how the magnetic field affects the mass transfer, and the mechanism is not definitively concluded. For example, many fellow researchers accredited the bubble movements to the direct Lorentz forces on the gas bubbles, with some of these works being quite recently published (e.g. *Scientific Reports*, 2016, **6**, 21068;). Please note that in the work cited by the reviewer, for which we now included a citation, the Lorentz force is explored in other reactions (CO₂RR) but the effect of the interaction of the Lorentz force with the bubbles is not clearly revealed, as mentioned by the authors in the conclusions "*We also identified two future directions for research [...]: 1) mechanistic exploration of how the Lorentz force interacts with electrochemical systems that produce bubbles*". Our work gives the direct evidence to the fact that the forces are applied on ionic species, and not directly on the bubbles, where the bubble movement is a secondary effect. To the best of our knowledge, this is the first conclusive study of this matter, and we believe it should be highly considered.

Other important issues:

1. A critical issue arises from the methodology employed: all the conclusions drawn by the authors lie on the utilization of phenolphthalein dye (H_2In , colorless) as a chemical indicator. This dye undergoes a color change to purple (In^{2-} , purple) in an alkaline when it interacts with OH^- ions. However, it's important to note that once phenolphthalein dye forms a complex with OH^- , the resulting purple species (In^{2-}) also carry a negative charge. Consequently, any nonparallel movement of these charged dye molecules, possibly due to concentration gradients, would subject them to the influence of the Lorentz force under a magnetic field. This implies that all the observed behaviors analyzed by the authors pertain to the charged dye In^{2-} , rather than the OH^- ions themselves. Furthermore, it is worth noting that some reports have suggested that OH^- ions may not physically migrate during electrochemical processes (e.g., proton hopping mechanism for OH^- ions "transfer").

We appreciate the reviewer's comment and the concern raised regarding the utilization of phenolphthalein dye (H_2In) as a chemical indicator in our study. However, we would like to mention that the utilization of phenolphthalein dye is not the only way for us to make meaningful conclusions. The experiments shown in Fig.2 and Fig.3a cannot be simply ignored. In fact, these results led to the conclusion that the Lorentz force acting on the moving ions is responsible for the enhancement of mass transport, not on the gas bubbles directly. The experiment using the dye was to further confirm our findings and an attempt of quantification of the effect.

It is indeed true that phenolphthalein undergoes a color change to purple (In^{2-}) in an alkaline environment when interacting with OH^- ions, and as a result, the resulting purple species (In^{2-}) carries a negative charge. However, we want to clarify that our conclusions do not hinge solely on the behavior of OH^- ions under a magnetic field, although we would like to highlight that we expect a net ionic current composed mostly of OH^- . Given the nature of the reaction, OH^- is generated at the WE and consumed at the CE. In any case, our study focuses on the response of all charged ions present in the electrolyte. These ions, whether OH^- or In^{2-} , when moving away from the WE, experience altered trajectories due to the electric field created by the applied potential and the additional influence of the Lorentz force when a magnetic field is applied. In this context, the distinction between tracking OH^- or In^{2-} ions becomes less relevant, as the fundamental observation is that charged ions in the electrolyte change their paths in response to the Lorentz force. The change in trajectory of the charged species, caused by the Lorentz force, leads to a stirring effect on all species in proximity to the electrode surface.

Furthermore, considering that the diffusion of OH^- is known to be significantly faster than that of organic anions in water, we can assume that the change in colour at the limits is mostly caused by the appearance of OH^- ions in the vicinity, rather than the diffusion of the indicator itself.

As far as we know, the exact mechanism of OH^- diffusion in bulk water is not fully understood. Theoretical approaches indicate that both Grotthuss and vehicular mechanisms are present, with different relative contributions (*J. Phys. Chem. B* 2017, 121, 6, 1362–1371). For example, the ratio between the diffusion coefficient estimated for each mechanism, D_G/D_V is 1.47 at 300 K but decreases to 0.61 at 360 K, indicating that both contributions are relevant at room temperature. Furthermore, the theoretical approaches do not consider any applied potential, which could also have an influence on the diffusion mechanism. In addition, high ionic strengths -as is the case in our experiments- can significantly modify the diffusion of OH^- (*J. Am. Chem. Soc.* 2019, 141, 17, 6930–6936).

With this in mind, while we cannot discard the presence of Grotthuss diffusion, it is evident from our experiments that there is a change in the path of charged ions in response to the Lorentz force, thus indicating that vehicular motion could be significant in this case. Furthermore, the estimated diffusion coefficient is consistent with a value reported for OH⁻, and quite larger than that of organic molecules, which could be an indication that we are actually following the OH⁻ movement.

We hope this clarifies the basis of our conclusions and the impact of magnetic fields on the motion of charged ions.

2. A lack of essential information regarding the applied magnetic field. Firstly, the direction of the magnetic field is not provided, as commonly understood, the Lorentz force acting on charged particles is direction dependent. The authors mentioned that "ORR measurements were conducted on a Pt foil parallel to the magnetic field," but there are two directions for parallel and they did not clarify the conditions for OER. Secondly, information of uniformity, strength gradient, distance between magnetic poles, and operating temperature is missing. Given that ORR is highly temperature-sensitive, the absence of details regarding the magnetic poles' temperature and the distance to the electrochemical cell raises concerns about the validity of the results.

We appreciate the reviewer's feedback regarding the essential information concerning the applied magnetic field, and we have taken steps to address these concerns:

- Direction of the magnetic field: We have added the direction of the magnetic field in Figure 3. The 'parallel' direction of the magnetic field in the experiments shown in Figure 6 is the same as that of Figure 3 d and further clarified in Suppl. Figure 1. OER was performed in the same configuration. Analogous results were obtained when the magnetic field was set to the opposite direction for both reactions, thus only one case is shown for clarity.
- Additional details about magnetic field: The magnet poles have a diameter of 10 cm, ensuring a uniform magnetic field between the two poles. We placed the working electrode (WE) with an area of less than 2 cm² in the centre of the poles to guarantee uniformity in its vicinity. The magnetic field was measured at the WE position with a Hall probe before each set of experiments. We have included this information in the Methods section and a more detailed scheme and photograph of the setup was added to the SI.
- Temperature: The ORR experiments were conducted at room temperature. The magnetic poles were water-cooled, and no noticeable heating was observed on the outside. The electrochemical cell was placed in the centre of the poles, with 0.5 cm of space on each side.
- Refer to Blumenschein et al. (*Phys. Chem. Chem. Phys.* 2020, 22, 997) for evidence of the magnetic field homogeneity and stability in another electrochemical experiment conducted using the same magnet as the one used in this study. EPR spectra of 1 mT in width were recorded in a special data-taking scheme required for that other experiment. Thus, data points were accumulated over a time span of several days. Thus the field homogeneity is better than 1 mT and its value is stable over this field range for days, while running an electrochemical experiment.

By including these additional details, we aim to provide a more comprehensive understanding of the experimental conditions and address the reviewer's concerns about the validity of the results.

3. The authors claimed that "a magnetic field-induced mechanical force is applied on the species

involved in the ORR, HER and OER (for instance, OH⁻), but not on the O₂ bubbles directly.” (Line 244-246). However, in the case of ORR, the reactants in the electrolyte are not charged (O₂, H₂O), but the charged OH⁻ ions is the resultant. This raises a critical question: is the observed increase in limiting current a result of the magnetic field enhancing the diffusion of resultant OH⁻ ions? What is the underlying reason for this? Although the ORR does not directly produce gaseous products, it does require oxygen gas (O₂) as a reactant. Given that the Lorentz force affects charged ions in the electrolyte, whether this force generates any secondary effects that indirectly influence the behavior of oxygen bubbles, and consequently the ORR process?

We would like to clarify that the observed increase in the ORR limiting current is not a direct result of the enhanced diffusion of the product (the resultant OH⁻ ions) but rather results from the collective stirring of the entire electrolyte which improves the availability of the reactant (dissolved O₂). This stirring is generated by the Lorentz force, acting on the moving charged species (towards or away from the WE, depending on their charge and applied bias). This allows O₂ molecules, which dissolve at the liquid/gas interface, to reach the working electrode (WE) more readily. In our understanding, the reactant in the ORR is not gaseous oxygen (O₂) but rather dissolved oxygen, which is affected by the movements of species in the electrolyte. The magnetic field-induced stirring facilitates the transport of dissolved oxygen from the liquid/gas interface to the working electrode (WE), making it more accessible for the ORR. It is not a direct influence on oxygen bubbles but rather an indirect effect on the availability of dissolved oxygen, similar to how stirring with a magnetic stirrer would promote better mixing in the electrolyte. We hope this clarifies the underlying mechanism for the observed increase in limiting current during the ORR. We have added this explanation to pages 20-21 of the manuscript.

4. The authors have not provided a clear explanation of the differential effects of mass transfer on the OER and ORR. The OER involves charged OH⁻ ions and produces O₂ gas, and the ORR involves O₂ gas and produces OH⁻ ions, but both OH⁻ and O₂ coexist in their electrolytes. If the Lorentz force induced by the magnetic field primarily acts on OH⁻ ions, but no improvement in the OER while an improvement in the ORR is observed, it raises questions that mass transfer of reaction products is important than that of reactants? Besides, as stated by authors that a secondary effect is imposed on O₂ bubble, but what is the difference for OER and ORR? Thus, further clarification about the effects on OH⁻ ions and O₂ bubbles for OER and ORR are needed.

We would like to note that the observed improvement in ORR while no significant change in OER is observed can be attributed to the specific characteristics of these electrochemical processes and the role of the Lorentz force.

As previously mentioned, the magnetic field-induced Lorentz force affects the trajectory of the ionic species moving in the electrolyte. This induces a stirring effect in the electrolyte, promoting mass transport. In the case of OER, the reactant typically involves the solvent (H₂O or OH⁻ in alkaline media), which is highly available, and changes in the mass transport of the electrolyte do not significantly impact the reaction current. The primary impact, if any, is on the removal of O₂ bubbles adhering to the electrode, although our results show that this effect is not very significant for OER.

On the other hand, ORR limiting current is strongly dependent on the availability of the reactant, which is dissolved oxygen near the working electrode (WE). The current-overpotential equation made by A.J. Bard, A. J. Bard and L. R. Faulkner, *Electrochemical Methods* (Wiley, 2000), clearly shows the importance of the concentration of both the reactants and the products:

$$i = i_0 \left[\frac{C_O(0, t)}{C_O^*} e^{-\alpha n f \eta} - \frac{C_R(0, t)}{C_R^*} e^{(1-\alpha) n f \eta} \right]$$

The reaction's performance can vary significantly based on the environment, such as air or O₂-saturated conditions. Similarly, the state of the electrolyte, whether still or stirred, has a substantial impact on the limiting current. Here, we demonstrate that the Lorentz force induced by the magnetic field affects the motion of ions in the electrolyte, leading to a stirring effect. This stirring effect is significant for the ORR, where the availability of dissolved oxygen plays a critical role. We hope this clarification highlights the underlying reasons for the observed differential effects between OER and ORR in the presence of a magnetic field. We have added this explanation to pages 20-21 of the manuscript.

5. The reliability of the observed increase in the ORR limiting current using this system is questionable. It is suggested to employ a standard Rotating Ring-Disk Electrode (RRDE) setup to enhance the credibility and consistency of the ORR tests. The RRDE configuration is widely accepted in electrochemical research for ORR testing as it allows for the separation of reaction products (such as H₂O₂), quantitative analysis, and reliable comparisons.

We agree that RRDE configuration is the standard technique used for ORR testing, as it removes mass transport effects among other advantages. However, in this case we aimed to explicitly observe mass transport changes during the electrochemical reaction, that would otherwise be avoided in a RRDE setup. RRDE and RDE are scientific instruments. The use of them helps the study of reaction kinetics. However, the use of these instruments or methodology of them, is not practical in real-life applications, for instance, water electrolyzers and fuel cells.

6. Pt is not suitable as a working electrode (WE) for OER testing as it can undergo oxidation on the anode, leading to the dissolution of Pt ions into the electrolyte solution. The charged Pt ions with a concentration gradient can also be influenced by magnetic field. Thus, the designed system with Pt as WE not only contaminate the electrolyte but also affect the accuracy and reliability of the results.

We appreciate the reviewer's concern regarding the potential issues related to Pt dissolution. We would like to provide a more comprehensive explanation for our choice of Pt.

Pt was selected as the WE for several reasons. Firstly, it is a widely accepted and commonly used noble electrode in electrochemical studies, which is also non-magnetic, a property essential to avoid any magnetic interaction besides the external magnetic field.

Additionally, while it is true that Pt can undergo oxidation during the OER, it is essential to recognize that all catalysts may experience some level of oxidation during the OER due to the nature of the reaction. Furthermore, when compared to many other materials, Pt exhibits one of the lowest dissolution rates (*Electrocatalysis*, 2018, 9, 153–161). Various studies have shown that other OER catalysts, such as iridium and ruthenium also undergo oxidation and dissolution during the OER. The dissolution of Ir and Ru is well-documented in the literature (*Inorg. Chem. Front.*, 2021, 8, 2900-2917 and *Catalysis Today* 2016, 262, 170-180). Other OER catalysts, for instance, first-row transition metal oxides, tend to have even higher dissolution rates compared to noble metals (*Nature Energy*, 2020 5, 222–230). Therefore, we believe Pt is the best choice for our studies of the effects on mass transport.

It's also important to emphasize that in Figure 3, where the key results supporting our interpretation of the mass-transfer mechanism are presented, Pt serves as the cathode, and no dissolution is expected.

Overall, we believe Pt is the best choice for our study on the effects of mass transport. Our findings are based on the behaviour of charged species in the electrolyte under the influence of the applied electric field and magnetic field, and we are confident that our conclusions remain valid and applicable.

7. The authors presented electrode currents, such as -0.2, -0.4, and +0.4 mA, without additional details. Raw currents lack meaningful interpretation, as they depend not only on the applied potential but also on the electrode area. It is suggested to normalize the currents by the electrochemical surface area (ECSA) or electrode area to enhance the clarity and relevance of these results. Additionally, it would be better to specify the techniques employed to measure these currents—whether CA or CP methods were used? The applied potential for each of these cases should also be explicitly stated to provide a comprehensive understanding of the experimental conditions.

We appreciate the reviewer's valuable input regarding the presentation of electrode currents and the need for additional details. We have taken these suggestions into consideration.

1. The electrode currents were controlled by performing chrono-potentiometry (CP), as was described in the Methods section: *"Videos were recorded through a hole in the electromagnet pole during chronopotentiometry control at different electrode currents (-0.2, -0.4 and +0.4 mA)"*. We have clarified this fact in the corresponding figures. An example of the CP obtained while recording Suppl. Video 3 was already included in the supplementary information (Suppl. Figure 10) to provide a comprehensive understanding of the experimental conditions.

2. While we agree that ECSA is important for the evaluation of catalytic materials, we did not normalize the currents by electrochemical surface area (ECSA) in Figure 3. We want to clarify that the results presented in Figure 3 are primarily related to the behaviour of charged species in the electrolyte. These results are not expressions of catalytic activity, and therefore, the actual ECSA may not be critical for our interpretation of these specific findings. In general, we have attempted to maintain the same current, i.e., charge flow, in the electrolyte to study the magnetic field effect on mass transport. We believe that the consistency in the overall current is more suitable than in the specific current density. On the other hand, it's worth noting that we have determined the ECSA for the ORR analysis in Figure 6, where it plays a more central role in the evaluation of catalytic activity.

8. It's important to clarify the basis for the depiction of the Lorentz force acting on the diffusion of OH⁻ ions in Figure 4. Is it derived from experimental observations or theoretical modelling? Firstly, as previously mentioned in point 1, there is no direct experimental evidence of OH⁻ diffusion presented in the study. Secondly, the work lacks the inclusion of theoretical models that could generate these results.

This representation is based on our interpretation of the experimental results, complemented by a simplified theoretical model describing the induced Lorentz force applied to moving charges in the electrolyte. In Figure 4, our aim was to present a simplified theoretical model where a charged species, which would typically move linearly toward the working electrode (WE) in the absence of a magnetic

field, follows a spiral trajectory due to the influence of the Lorentz force. We have clarified the description of the phenomena.

As previously mentioned, we did not intend to imply that only OH⁻ ions are affected by the Lorentz force. Instead, our interpretation is that all ions in motion within the electrolyte are influenced. OH⁻ ions were selected to illustrate the concept, because during OER it is most likely that OH⁻ are depleted near the WE and thus new OH⁻ from the bulk are diffusing towards the electrode. We appreciate the reviewer's feedback as it has helped improve the clarity of our presentation.

9. As per the authors' exposition, the flat Pt foil and Pt mesh manifest notably disparate OER behaviors, as evidenced in Figure 6a and b. The onset potential for the flat Pt foil is significantly diminished in comparison to the Pt mesh. It would be beneficial to elucidate the underlying reasons for this disparity. Additionally, a comprehensive explanation regarding how the geometrical configuration of the electrodes modulates mass transfer in the presence of a magnetic field is desirable.

We appreciate the reviewer's concern and suggestions. We would like to clarify the differences observed in the Figure:

- The Pt mesh experiment in Fig 6.b was conducted in an O₂-saturated electrolyte. Compared to Fig 6.a, an increase in the OER overpotential is expected. In any case, we also conducted the experiments with Pt mesh in air, and we modified the figure to include this data instead, to avoid any confusion by the different experimental conditions. As the reviewer can see, the OER current is not evidently affected in either case.
- As different samples are fabricated differently, the ohmic resistance, e.g., contact resistance between the electrode and the current connector, cannot be maintained at the same value for all samples. The results are not IR-corrected. Therefore, we primarily report comparisons in the changes of the reaction current within the same sample.

We agree with the reviewer that a comprehensive explanation regarding the geometrical-related mechanisms is very beneficial. However, we would like to stay focused on the effects on mass transport in the electrolyte, especially figuring out the mechanism and which electrolyte substances are directly involved. Further studies on geometrical configuration are indeed essential to the application of the magnetic effects but are beyond the scope of this communication.

10. Referencing Supplementary Figure 7, it's evident that magnetic fields exert a tangible effect on adsorptive behaviors exhibited on electrode surfaces. This could potentially correlate with alterations in the bilayer structure of the electrode surface, as underscored by prior works (J. Phys. Chem. B, 2001, 105, 9487–9502; J. Phys. Chem. B, 2004, 108, 5778–5784).

We appreciate the reviewer's suggestion and the papers provided here, and we have included the relevant citations to the manuscript. We agree that the magnetic field affects the double-layer structure on the electrode surface during the reaction. It is indeed possible that the magnetic field changes the thickness of the diffuse layer. As we mentioned above, the focus of this manuscript is to demonstrate the magnetic effect on the mass transport of electrocatalytic reactions and reveal the

actual mechanism of this effect. We do think what the reviewer pointed out here is worth a stand-alone study in the future.

Reviewer #3 (Remarks to the Author):

Authors investigated the effect of magnetic field on mass transfer by means of optical vision and electrochemical testings. Authors have made some progress compared to the reported references, more clearly demonstrated mass transport in the magnetic field. However, the innovation of the article needs to be discussed. Some comments are as follows,

1. What are highlights in this manuscript, which innovations make differences from other references such as Scientific reports 6 (1), 21068 (2016); Sci Rep 11, 9346 (2021); ACS Applied Energy Materials 3 (7), 6752-6757 (2020).

We appreciate the reviewer's interest in our work. We believe our study introduces several unique contributions and innovations that sets it apart, as we describe here. We have also highlighted the novelty in the abstract and introduction (pages 4-5).

1. Direct visualization of Lorentz force effects in the electrolyte: Unlike the references mentioned, which mainly rely on the observation of bubble behaviour during OER, our study focuses on the direct visualization of the motion of charged ions under the influence of a magnetic field. Our findings provide a more comprehensive understanding of the effects of the Lorentz force on electrocatalytic processes and shed light on the cause of rotational movement.

Previous research has suggested various explanations for the motion of bubbles during electrochemical reaction with applied magnetic fields, such as:

- (i) the Lorentz force is applied to the bubbles due to the presence of a surface charge, caused by adsorbed ions at the bubble-electrolyte interface (e.g. *J. Electrochem. Soc.* 2018, 165, E679),
- (ii) as bubbles rise to the surface due to the buoyance force, anions adjacent to them are dragged by this motion, resulting in an ion current, which is then affected by the magnetic field (e.g. *Scientific reports* 2016, 6 (1), 21068; *ACS Appl. Energy Mater.* 2020, 3, 7, 6752–6757), or
- (iii) the evolving paramagnetic O₂ bubbles are deflected due to the interaction with the magnetic field (e.g. *Scientific reports*, 2021, 11, 9346).

Our research, however, offers a different perspective. We demonstrate that the primary factor driving the phenomenon is the Lorentz-force-induced movement of ions dissolved in the electrolyte, which induces a general stirring of the solution. We show that bubble movement is a secondary effect and not the primary driver, as previously hypothesized. According to the results shown in this manuscript, we believe our interpretation is more accurate.

2. Universality of this effect: The mentioned references are centered specifically on water electrolysis. Our study explores the broader relevance of magnetic field effects on mass transport, which extends beyond particular reactions and offers a universal perspective on this phenomenon. We explored the implications of the Lorentz force in various electrochemical contexts, shedding light on its importance for diffusion-limited reactions, and offering new opportunities for developing more efficient and sustainable energy conversion technologies.

2. The mechanism of mass transfer is not clear, please adding simulation calculations.

We appreciate the reviewer's suggestion. While simulation could provide valuable insights into mass transfer processes, we are not aware of any simulation methods that could currently allow us to model the inherent complexity of the interactions between charged species and the Lorentz force in a magnetic field within an electrochemical system.

We have described the mass transfer effect on the ions in more detail in the text (page 13) and in Figure 4, and we summarize it here: During the electrochemical reactions, an ionic current emerges due to the depletion of reactants at each electrode. In the OER in alkaline conditions, this ionic current mostly comprises OH⁻ ions. The trajectory of these ions is changed by the appearance of the Lorentz force. Considering that this force is perpendicular to both the ion's velocity and the magnetic field direction, this induces a rotational movement of the anions. In the case of the HER and ORR, the direction of the ionic current is reversed. Thus, the Lorentz force also reverses its direction, leading to the formation of a vortex in the opposite sense.

REVIEWER COMMENTS

Reviewer #1 (Remarks to the Author):

“1) During the decomposition of H₂O₂, how to ensure that no charged particles are generated on the surface of O₂ bubbles? If a thin tube is used to blow bubbles into acidic, alkaline, and neutral deionized aqueous solutions, will the same conclusion still be obtained under the action of external magnetic field?”, maybe my first comment wasn't clear enough.

The first experimental suggestion is to distinguish between the directional movement of charged particles under an electric field and the diffusion transfer due to the ion concentration gradient caused by a chemical reaction, because they both have the potential to cause the transverse motion of bubbles in a solution of charged particles. Through the difference of bubble movement behavior in deionized water and alkaline (acidic) solution, the real picture reflected by the change of bubble movement trajectory can be further clarified. Therefore, I would like to suggest the authors to further confirm your claims with experiments, and then consider whether to publish them in NC.

Reviewer #2 (Remarks to the Author):

The effort the authors have invested in improving the manuscript should be appreciated. While after careful consideration of the revised manuscript and the responses provided to the previous round of reviews, I think that it is unable to accept the manuscript for publication in its current form.

The decision is primarily based on the observation that the revisions made did not sufficiently address the concerns raised by the reviewers during the previous evaluation. As a peer reviewer, it is essential for me to ensure that the manuscript meets the rigorous standards set by Nature Communications, and unfortunately, the current version falls short of these expectations.

I encourage the authors to carefully consider the reviewers' comments and make substantive revisions to address their concerns comprehensively.

Reviewer #3 (Remarks to the Author):

Authors investigated magnetic field effect on OER/ORR, attaining the phenomena about bubbles movement by way of optical instrument. The following comments are required to be stated.

1. Inaccurate understanding about "the Lorentz force acting on the electrolyte ions induces a vortex-type motion", which may be related to testing device?

2. What are the electrolytes? the three mass transport modes simultaneously occur in the electrochemical reaction. "Since mass transport in electrocatalysis mainly comprises diffusion, migration or convection of ionic species in the electrolyte, the effects of the magnetic field on the electrolytes require further investigations and evaluation."

3. The magnetic field on/off and the distance factors are used for qualitatively/quantitatively analyzing the magnetic field effects, which are not enough for magnetic field intensity.

4. The magnetic field can promote HER/ORR/OER demonstrated in other references, this manuscript is further focused on expounding highlights and distinctive insights.

Reviewer #1 (Remarks to the Author):

"1) During the decomposition of H₂O₂, how to ensure that no charged particles are generated on the surface of O₂ bubbles? If a thin tube is used to blow bubbles into acidic, alkaline, and neutral deionized aqueous solutions, will the same conclusion still be obtained under the action of external magnetic field?", maybe my first comment wasn't clear enough. The first experimental suggestion is to distinguish between the directional movement of charged particles under an electric field and the diffusion transfer due to the ion concentration gradient caused by a chemical reaction, because they both have the potential to cause the transverse motion of bubbles in a solution of charged particles. Through the difference of bubble movement behavior in deionized water and alkaline (acidic) solution, the real picture reflected by the change of bubble movement trajectory can be further clarified. Therefore, I would like to suggest the authors to further confirm your claims with experiments, and then consider whether to publish them in NC.

We thank the reviewer for the question. We have made several observations that allow us to show that both the magnetic field and the electrochemical reaction need to be present for the transversal movement of the bubbles to occur:

- 1- The clearest demonstration of this fact is the sudden change in the bubble movement observed in the last seconds of Supplementary Video 3 (previously named Suppl. Video 5 (*right*)): When the potential is applied, the bubbles follow a whirling movement in one or other direction depending on the direction of the magnetic field. When the application of potential is stopped (time stamp 1:11:75), the bubbles suddenly stop moving laterally and move upwards with their natural motion, even though the magnetic field is still being applied. If the bubbles were superficially charged, this sudden change would not be expected. This would also not be expected if the bubbles movement was caused by displacement of ions in their surroundings. This was highlighted in the text in page 7: "*We noted that the force on the gas bubbles vanished immediately when the reaction was turned off, although the magnetic field remained (see last seconds of **Suppl. Video 3**). Therefore, we can conclude that the reaction current is essential to the generation of the forces on the bubbles.*"
- 2- The second observation are the experiments performed with H₂O₂: To distinguish between the effect of the applied magnetic field in the bubble movement, or other phenomena that can have the potential to cause the transverse motion of bubbles, we performed the experiments without the magnetic field (as shown in **Figure 2a**, lower panels). Here, we observed that even with a current of -25 mA on the Au electrode, no deflection of the bubbles takes place. This indicates that neither the electric field caused by the applied potential, nor the diffusion gradient caused by consumption of ions due to the electrochemical reaction, generate a force strong enough to change the direction of the bubbles evolving from the Pt wire. The deflection occurs only when the magnetic field is applied (**Figure 2a**, top panels). This explanation was included in the main text (page 9): "*Additionally, the bubble stream was not disturbed in the absence of magnetic field, regardless of the current applied at the Au coil, indicating that neither the electric field - caused by the applied bias- nor the concentration gradient -caused by the electrochemical reaction- have an influence in the bubble stream direction.*"

We observed this behaviour of the O₂ bubble movements during OER both in basic and neutral electrolytes, which presented the same behaviour in all cases. Additionally, the dye experiments allowed us to visualize the change in the movement of the ions in solution, further helping to verify our hypothesis: that the Lorentz force is acting on the ions that are moving due to the electrochemical reactions. We hope this clarifies the basis of our observations.

Reviewer #2 (Remarks to the Author):

The effort the authors have invested in improving the manuscript should be appreciated. While after careful consideration of the revised manuscript and the responses provided to the previous round of reviews, I think that it is unable to accept the manuscript for publication in its current form.

The decision is primarily based on the observation that the revisions made did not sufficiently address the concerns raised by the reviewers during the previous evaluation. As a peer reviewer, it is essential for me to ensure that the manuscript meets the rigorous standards set by Nature Communications, and unfortunately, the current version falls short of these expectations.

I encourage the authors to carefully consider the reviewers' comments and make substantive revisions to address their concerns comprehensively.

We have made our best effort to thoroughly answer all the detailed concerns posed by the previous reviewers and included any necessary explanation throughout the revised text. In response to the issues raised, we have diligently revised the manuscript, incorporating detailed explanations and modifications throughout the text. A summary of the changes made in response to each point raised by the previous reviewers is provided below. Out of 17 comments, only 2 (questions 5 and 7 from Reviewer #2) did not result in an addition or modification of the original text, as these comments proposed a change in methodology that was out of scope of our work (choice of EC-cell, and choice of Pt as electrode). If the reviewer could provide more specific details on the concerns they believe were not sufficiently addressed, we would be more than willing to further enhance the manuscript. We remain committed to ensuring the highest quality and relevance of our work and are open to any additional guidance or suggestions. Notwithstanding, we are also aware of the relevance and timeliness of this work and we believe that the thorough corrections made so far permit to reach a swift decision.

Reviewer #1

- 1) We added a clearer explanation on the effect of the magnetic field in the bubbles formed by H_2O_2 decomposition in pages 8-9 of the manuscript.
- 2) We improved our description of the magnetic field effect in pages 20-21.
- 3) We clarified the experimental background colours in the Methods section.
- 4) We clarified the experimental conditions of the electromagnet in the Methods section and included Suppl. Figure 1.
- 5) We added further experimental evidence of the situation on the left side of the WE to the SI.

Reviewer #2

- 1) We improved our description of the observations with the dye in page 13.
- 2) We added additional details on the electromagnet setup in the Methods section. We also clarified the direction of the magnetic field in Figure 3.
- 3) We gave a more detailed explanation in pages 20-21.
- 4) We gave a more detailed explanation on the difference in the effect of the magnetic field in OER and ORR in pages 20-21.
- 5) We did not feel the need to clarify in the text why we didn't use a RRDE.
- 6) The choice of Pt as non-magnetic electrode was previously described in the manuscript. We do not think it is necessary to add a discussion on the dissolution of catalysts during OER as this is not the topic of this work.
- 7) We added details on the chrono-potentiometry conditions in the Methods section.

- 8) We have clarified the description of the phenomena caused by the Lorentz force in page 13 and improved Figure 4.
- 9) We modified Figure 6.b to have a more consistent comparison between two electrodes with different geometries in the same experimental conditions.
- 10) We included the suggested citations to the manuscript.

Reviewer #3

- 1) We highlighted the novelty in the introduction (pages 4-5) and included the suggested references.
- 2) We have clarified the mass transfer effect on the ions in more detail in page 13 and improved Figure 4.

Reviewer #3 (Remarks to the Author):

Authors investigated magnetic field effect on OER/ORR, attaining the phenomena about bubbles movement by way of optical instrument. The following comments are required to be stated.

1. Inaccurate understanding about "the Lorentz force acting on the electrolyte ions induces a vortex-type motion", which may be related to testing device?

We thank the reviewer for the comment. To avoid confusion, we removed the term "vortex" in the text and replaced it by "stirring" or "whirling motion". We would like to anyhow clarify that we did observe a whirling motion of the bubbles and the electrolyte during our experiments, which was schematized in Figure 4. Focusing in the area around the electrode, the ions that would naturally move straight towards (or away from) the electrode surface, are influenced by the Lorentz force which is perpendicular to the ion velocity and the magnetic field directions. This interaction induces a movement of the anions which is in any situation rotational (as described in page 13). Further away from the electrode surface, the movement would also be influenced by other interactions (i.e, reduced flow near the walls of the EC-cell). Direct observations of these whirling movements can be seen in the Supplementary Videos 2-4. We have clarified this in the text, as follows (page 7): "*In addition, experiments with Pt mesh show a whirling motion of the bubbles during the OER in the presence of the magnetic field (Suppl. Video 2). We observed that changing the magnetic field direction with the electromagnet (Suppl. Video 3) or locating a permanent magnet below the electrode (Suppl. Video 4) lead to the bubbles curling in different directions.*"

2. What are the electrolytes? the three mass transport modes simultaneously occur in the electrochemical reaction. "Since mass transport in electrocatalysis mainly comprises diffusion, migration or convection of ionic species in the electrolyte, the effects of the magnetic field on the electrolytes require further investigations and evaluation."

We thank the reviewer for the remark. The electrolytes in this case refers to any ionic species dissolved in the solution. We have clarified the sentence in the introduction as follows: "*Since mass transport in electrocatalysis mainly comprises diffusion, migration or convection of ionic species in the electrolyte, the effects of the magnetic field on **the movement of these species** require further investigations and evaluation.*"

3. The magnetic field on/off and the distance factors are used for qualitatively/quantitatively analyzing the magnetic field effects, which are not enough for magnetic field intensity.

We thank the reviewer for the remark. We are using an electromagnet, not a permanent magnet as previous examples from the literature. The magnetic field is homogeneous and the intensity at the position of the WE was determined with a Hall probe. We added detailed information of the magneto-electrochemical setup in the Methods section and Suppl. Figure 1 to clarify this aspect.

4. The magnetic field can promote HER/ORR/OER demonstrated in other references, this manuscript is further focused on expounding highlights and distinctive insights.

We thank the reviewer for the comment. Examples were cited from the literature of previous experiments enhancing OER and HER under magnetic fields (Refs. 3, 5-16, among others), while for ORR no experimental studies have been presented to our knowledge. Here we provide a rationale for the mechanism and a guidance for the conditions under which we can reach reproducible results. This is indeed a key issue in the field, as the advancement so far was very much on hold due to the lack of reproducible protocols. In addition, we aimed at maximizing the use of magnetic fields in general to enhance different electrocatalytic reactions. We have added the following sentences to the abstract and conclusions to highlight these facts:

In page 1: *“Therefore, when detecting an enhanced catalytic activity at a magnetic electrode, the effect of magnetic fields on mass transport must be assessed.”*

In page 22: *“In previous works, the magnetic field was shown to promote HER and OER, but the reason behind this enhancement is not yet completely understood. In this work, we distinguish between kinetic and mass transport effects. We focus on the magnetic field effects on the mass transport in energy-related electrocatalytic reactions using non-magnetic electrodes. We provide distinctive insights on the extent of the possibility of using magnetic fields for mass transfer enhancement and offer an explanation for its mechanism. Our results demonstrate that the magnetic field-induced Lorentz force on the moving ionic species is essentially responsible for a rotatory motion in the electrolyte that facilitates mass transport, which can almost double the diffusion of ions toward one side of the electrode. Understanding the role of the mass transport effects is essential to the reproducibility of magnetically enhanced electrochemical reactions.”*

REVIEWERS' COMMENTS

Reviewer #1 (Remarks to the Author):

Although the authors did not redesign the new experiment as suggested by the reviewers, they reasonably explained the reviewers' concerns in detail through the specific phenomena in the existing experimental data. If the author can comprehensively answer the concerns of all reviewers, it is considered that it can be further improved on the basis of the existing version and published in NC.

Reviewer #2 (Remarks to the Author):

After a careful consideration of the response letter and the revised manuscript, I think that the authors have made a good revision according to all reviews's suggestion and comments. I would like to recommend it to be accepted.

Reviewer #3 (Remarks to the Author):

This paper comprehensively investigate the effect of magnetic field on mass transfer of oxygen redox reactions, which provides the deep insight for electromagnetic coupling. In view of the right comments, I am able to recommend this paper to be published in Nature Communications.